# Google Earth Engine Framework for Satellite Data-Driven Wildfire Monitoring in Ukraine

Bohdan Yailymov [1], Andrii Shelestov [1,2], Hanna Yailymova [1,2,*] and Leonid Shumilo [3]

1 Department of Space Information Technologies and Systems, Space Research Institute NAS Ukraine & SSA Ukraine, 03187 Kyiv, Ukraine; yailymov@ikd.kiev.ua (B.Y.); andrii.shelestov@lll.kpi.ua (A.S.)
2 Department of Mathematical Modelling and Data Analysis, National Technical University of Ukraine 'Igor Sikorsky Kyiv Polytechnic Institute', 03056 Kyiv, Ukraine
3 Department of Geographical Sciences, University of Maryland, College Park, MD 20742, USA; lshumilo@umd.edu
* Correspondence: yailymova.hanna@lll.kpi.ua

**Abstract:** Wildfires cause extensive damage, but their rapid detection and cause assessment remains challenging. Existing methods utilize satellite data to map burned areas and meteorological data to model fire risk, but there are no information technologies to determine fire causes. It is crucially important in Ukraine to assess the losses caused by the military actions. This study proposes an integrated methodology and a novel framework integrating burned area mapping from Sentinel-2 data and fire risk modeling using the Fire Potential Index (FPI) in Google Earth Engine. The methodology enables efficient national-scale burned area detection and automated identification of anthropogenic fires in regions with low fire risk. Implemented over Ukraine, 104.229 ha were mapped as burned during July 2022, with fires inconsistently corresponding to high FPI risk, indicating predominantly anthropogenic causes.

**Keywords:** wildfire monitoring; burned area mapping; normalized burn ratio; fire potential index; Google Earth Engine; informational technology; cloud computing

## 1. Introduction

In recent years, the degradation of the environment due to widespread fires has been observed [1–3]. This highlights the importance of detecting and monitoring forest fires, as well as effectively managing their spread. Satellite data has proven to be an effective solution for addressing this issue.

Because of the war in Ukraine, which began on 24 February 2022, we need to receive information with a higher spatial resolution than, for example, in Europe, to better understand the risks of fire occurrence, especially in active military zones, and there is also an important question regarding the analysis of the impact of military actions on the occurrence of fires. More than 15% of the territory of Ukraine is covered by forest, but an equally important ecosystem is agriculture (more than 40% of the entire territory of Ukraine), which suffers greatly as a result of fires.

Unfortunately, for the territory of Ukraine, the monitoring system of meteorological data measurements is very poorly developed, which does not provide an opportunity to fully repeat the practice of the Canadian and US systems. But in Ukraine, research has been started on wartime fire monitoring using available data sources. In particular, report [4] focuses on assessing fire occurrence and its relationship to armed conflict in Ukraine using VIIRS Active Fire data and analyzing the impact of armed conflict on gross primary productivity (GPP). The authors showed that there is no significant correlation between conflict and fires at the country level, but when examining specific regions, a clear connection between conflict events and the occurrence of fires is revealed. In their work, an important emphasis is placed on taking into account weather and climatic conditions.

Fires can occur in unexpected locations, and can be caused by a variety of factors, including human activity, lightning strikes, and other natural causes. Since military operations are taking place in the southern and eastern territories of Ukraine, more attention is devoted to them. The authors of paper [5] show the possible use cases based on remote sensing data to track and monitor Ukraine's environment during the war. However, the authors do not provide any quantitative assessments at the country level, and present only the monitoring of small areas (pilots).

The Ukrainian Hydrometeorological Institute has also developed a fire monitoring system in 2020 [6]. It utilizes open data from NASA satellite infrared sensors to geocode thermal points and identify the origin, severity, area, and other characteristics of wildfires. The authors review the use of the fire monitoring system to track wildfires resulting from hostilities. Major criteria for identifying combat-caused fires are defined, including sudden outbreaks in unusual locations such as urban areas, irregular fire contours, diverse land use areas, unusual timing, and high severity. The research represents the first real-time wildfire monitoring using satellite data during warfare, enabling the identification of essential features of these fires. The disadvantage of this approach is low spatial resolution and, as stated in article [7], the use of such data underestimates the number of identified fires by 30–60%. In addition to establishing the locations of fires, the determination of the cause of the fire—due to dry weather conditions [8,9] or due to human activities—is currently also an urgent issue at the country level. It is especially important during the war in Ukraine when some territories are not accessible.

Globally, it is believed that the main causes of fires are anthropological or natural factors [10]. That is why our study addresses two research questions:

(1)　Where have fires occurred?
(2)　What is the reason of fires? Is it caused by meteorological or anthropological conditions?

To address the first question, we propose to use fusion of high (10 m) and lower (1000 m) data for burned areas detection. Since most of the open systems discussed above can only identify potential fires, more research is needed to confirm whether a detected hot spot is actually related to an active fire. And for this, we offer a framework (information technology) that consists in combining temperature hotspots with low-resolution FIRMS data and monitoring active fires and burned areas within them with high-resolution Sentinel-2 data. In this way, we save time and computing resources, which would have been spent on searching for fires in those areas where there were none. In this case, we selected MODIS and VIIRS data due to higher resolution compared to analogues (for example, SEVIRI (Spinning Enhanced Visible and InfraRed Imager) with a resolution of about 4 by 6 km [11]).

To address the second research question, we analyze fire risks and meteorological conditions and determine the areas where fires are unlikely to occur due to weather. To solve this issue, we use the Fire Potential Index (FPI) [12], which makes it possible to predict the territories where fires have a reasonable probability of occurrence. The authors of work [13] used the FPI index to predict fires in the mountains in the Free State Province in South Africa and showed an overall accuracy of 89% of matches with real fires, and in work [14], the authors used three modifications of the FPI index and showed that at least two of them demonstrate an accuracy of 60% in relation to the occurrence of real fires.

Considering that the territory of Ukraine is one of the largest in Europe (see Section 3) and it is necessary to process a large amount of satellite data, there is a need to use a cloud platform. The Google Earth Engine (GEE) cloud platform is free and can be useful for solving this problem [15]. GEE provides access to a vast collection of satellite imagery from various sources, including Landsat, Sentinel, MODIS, and more, as well as allows for efficient processing of large-scale geospatial data in cloud servers [16].

The methodology presented in studies [17,18] provides access to regularly updated Sentinel-2 satellite imagery from 2015 to the present, as well as MODIS FIRMS data and Landsat-8,9, enabling the creation of high-precision products. The section "Data" describes the data sources that are used in the GEE platform for the framework implementation.

From the reviewed sources, it can be concluded that there are ready-made methodologies for monitoring active fires and burned areas, as well as systems for monitoring fire danger. Nevertheless, universally accepted technologies for determining the cause of a fire at country level are still lacking, and we aim to address this issue in this article. The following sections describe the existing fire danger monitoring systems in the world, as well as the data that were used to create the methodology for Ukraine including satellite data, provide a general scheme of the proposed information technology and the framework for its implementation on the GEE platform, and provide the obtained results of searching for burned areas and assessing fire danger levels for the entire territory of Ukraine. A comprehensive analysis of the received fires for 2021–2022 and the fire hazard map is conducted, and appropriate conclusions are drawn regarding the causes of fires.

## 2. Existing Fire Danger Monitoring Systems in the World

Numerous successful techniques for fire monitoring have been developed globally, and one of the most famous is the national system for rating the risk of forest fires—the Canadian Forest Fire Danger Rating System (CFFDRS) [19]. The CFFDRS has been under development since 1968, and currently, two subsystems—the Canadian Forest Fire Weather Index (FWI) System [20] and the Canadian Forest Fire Behavior Prediction (FBP) System [21]—are being used extensively in Canada and internationally. The calculation of the FWI system components is based on:

1. time series of daily meteorological observations (temperature, relative humidity, wind speed, and 24 h precipitation)—weather data from approximately 2500 stations in Canada and the United States;
2. elevation data—the elevation grid is derived from the US Geological Survey (USGS) with spatial resolution of 1 km;
3. the fuel types—the map of fuel types is derived primarily from forest attribute data [22] based on satellite imagery acquired by NASA's Moderate Resolution Imaging Spectroradiometer (MODIS—TERRA and AQUA). This map contains information about the forest type or grassland with the moderate resolution.

This system is tested and works well for Canada. Its disadvantage is low spatial resolution due to the use of meteorological data with a resolution of 1 km. In article [8], the authors modified the FWI forest fire danger index for the territory of Ukraine by increasing its accuracy based on the use of satellite data with a higher spatial resolution, as well as by using soil moisture deficit, in addition to the six sub-indices of the FWI system.

In 2007, after a five-year test phase that involved the implementation of various national fire danger indices in European Forest Fire Information System (EFFIS) [23], the EFFIS network adopted the Canadian Forest FWI System to harmonize the assessment of fire danger levels across Europe. The EFFIS publishes two indicators that provide information on the spatial/temporal variability of FWI compared to a historical series of about 30 years. The FWI is calculated from the European Centre for Medium range Weather Forecasts (ECMWF) model (with spatial resolution of 8 km and 1–9-day forecasts) [24], and the MeteoFrance model (10 km and 3-day forecasts for Europe). Also, since 2003, the following sources of forest fire risk assessment were considered by EFFIS [25]: meteorological fire risk [26], vegetation stress fire risk [27] and Fire Potential Index (FPI) [28]. The FPI model was introduced in [12] and was later revised as part of the Wildfire Assessment System (WFAS) to adapt to the greener conditions in the eastern U.S. during the summer. The original algorithm and its revised version were validated in California and Nevada, where correlations between FPI and fire locations were found to be significantly high [12].

Another well-known fire monitoring system is the United States Forest Service—Wildland Fire Assessment System, WFAS [29]. Currently, WFAS is based on data from weather observations conducted at 1500 fire weather stations throughout the United States and entered into the Weather Information Management System (WIMS) [30]. In addition to current and previous weather data, fuel types, as well as live and dead fuel moisture are used to calculate the fire danger level. However, fire hazard levels are measured only

at discrete points where the station is located. The estimation of values between stations is accomplished using an inverse distance-squared technique on a 10 km grid. While this method proves effective in regions with relatively dense station distribution, it does reveal certain limitations in areas with fewer stations. In particular, this approach does not work for the territory of Ukraine due to the lack of a sufficient number of ground stations.

The NASA Fire Information for Resource Management System (FIRMS) [31–33] detects active fires and thermal anomalies based on MODIS data with a 1 km spatial resolution, and Visible Infrared Imaging Radiometer Suite (VIIRS aboard Suomi NPP and NOAA-20) [34–36] delivers this information in near-real-time to decision makers. The VIIRS M bands and the day/night band have 16 detectors per scan (750 m spatial resolution per pixel), while the I bands have 32 detectors per scan (375 m resolution per pixel). However, the system does not guarantee that a detected hotspot is indicative of an ongoing fire. The advantage of MODIS and VIIRS data is their capability to provide near-real-time monitoring of fire propagation, with data updates occurring every 3 h. This information can also be accessed through other NASA resources such as WorldView [37]. Another frequently utilized data source is the SPOT-VGT data from the SPOT satellite [38], although its low spatial resolution poses a disadvantage.

In summary, the considered global systems make it possible to predict the most fire-danger areas using meteorological indicators, fuel moisture indicators and the land cover type. This makes it possible to assess the situation at the country or continent level. With the use of satellite data with a higher resolution, we are able to determine the exact burned areas on a regular basis and quantify the damage caused.

For the operational use of all this information, we need a framework where all the data would be collected and a forecast of fire danger would be proposed, as well as computing resources for calculations and data processing.

Satellite data at higher spatial resolution are used for a more accurate location and mapping of active fires. In particular, papers [39,40] present an algorithm that utilizes Landsat data to identify burned areas, applicable wherever Landsat data are available [41]. The algorithm generates burn probability surfaces using band values and spectral indices, and burn classifications are produced through pixel-level thresholding combined with a region growing process. Landsat Burned Area Essential Climate Variable products were generated for the conterminous United States from 1984 to 2015, which mapped a greater burned area compared to existing datasets such as the Global Fire Emissions Database (GFED) and Monitoring Trends in Burn Severity (MTBS).

Fires at the early stage require the use of satellite data at very high temporal resolution (e.g., SEVIRI [11,14]) to be promptly identified. On the other hand, Sentinel-2 MSI data may enable an efficient mapping of small fires every 5 days when adequate detection methods are used [42–44]. However, due to the high spatial resolution, collecting and storing data for an entire country's territory over an extended period can be challenging and resource-intensive in terms of processing requirements. Article [45] presents the initial findings of a study that investigated the use of spectral indices to assess the severity of a forest fire that occurred on Madeira Island in August 2016. At these stages, approaches to monitoring burnt areas were based on vegetation indices and screening of necessary areas based on the threshold value of these indices.

In recent years, there has been a growing interest in utilizing satellite imagery and machine learning-based approaches for wildfire detection and monitoring. In article [46], the authors present a method for mapping forest fire susceptibility using ensemble models based on the locally weighted learning algorithm. Ghosh et al. [47] propose a novel hybrid deep learning model that combines a convolutional neural network (CNN) and recurrent neural network (RNN) for forest fire detection, achieving high classification accuracy and outperforming existing state-of-the-art results.

There is currently a substantial body of literature on the use of satellite data and deep learning techniques for fire monitoring, as evidenced by numerous studies [48–50]. While deep learning approaches offer high accuracy with large amounts of data, index-based

methods have certain advantages over deep learning methods for fire detection using satellite data, including simplicity and ease of implementation, the ability to be applied to various types of vegetation cover and land use without the need for specific training datasets, and greater interpretability due to their physical meaning, which can aid in understanding the cause of fire occurrence.

## 3. Study Area

The study area for this research is Ukraine, the second largest country in Europe. Located in Eastern Europe, Ukraine encompasses an area of approximately 603,628 square kilometers with geographic coordinates between latitudes 44° and 53° N and longitudes 22° and 41° E [51]. Ukraine's diverse landscape comprises forests, grasslands, wetlands, and mountains. The Carpathian Mountains lie in the west, while the Crimean Peninsula is located in the south. The country has a temperate continental climate characterized by hot summers and cold winters.

In this study, fire identification was conducted during the summer months of 2022. The MODIS data were used to identify all fire hot-spots for the entire territory of Ukraine. For the areas where military operations are taking place (Figure 1, yellow regions—Kharkivska, Luhanska, Donetska, Dnipropetrovska, Zaporizka, Khersonska, and Mykolaivska oblasts), the fire boundaries were specified for 15–17 July 2022 based on Sentinel-2 data, and the areas of damage were determined by different land cover types. Special attention was paid to the natural ecosystem area near the National Nature Park of nationwide significance "Oleshkivski pisky", Kherson region (Figure 1, green box), where approximately 1.5 thousand hectares were burned in August 2022.

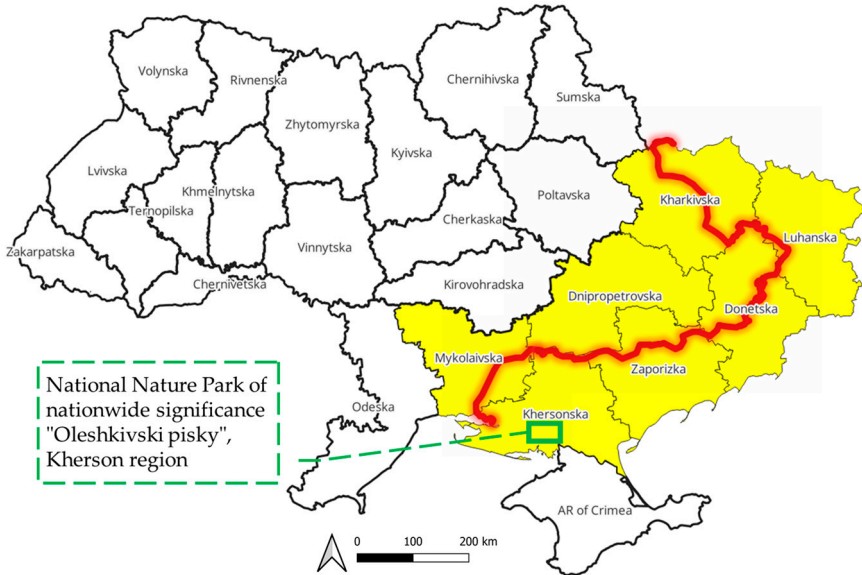

**Figure 1.** Ukraine—area of interest for determining burnt areas. The territories that are in the vicinity of active hostilities are marked in yellow. Red color—the front line and military operations as of August 2022.

In this work, special attention is given to forested areas, as their restoration requires a long time. Ukraine's forest covers an area of 10.4 million hectares, with 9.6 million hectares covered by forest vegetation [52]. The restoration of forests is crucial, as they play a vital role in the ecosystem, provide habitats for wildlife, help regulate the climate, and protect against soil erosion. Forest fires, particularly in the summer months, pose a significant threat to the environment, human health, and property damage, making it important to identify and monitor fires in forested areas.

Overall, Ukraine's diverse landscape and the ongoing military operations in the eastern and southern regions present a challenging environment for fire monitoring and

detection. The focus on the forested areas and the consideration of various types of land cover helps provide a better understanding of the extent and impact of fires in Ukraine, which can assist in developing effective strategies for fire management and prevention.

## 4. Data Used

The data used in this work with collection's links in Google Earth Engine (GEE) cloud platform and spatial resolution specification are presented in Table 1.

**Table 1.** Satellite data used in this article.

| Data | Spatial Resolution (m) | Collection in Google Earth Engine |
|---|---|---|
| MODIS (FIRMS) | 1000 | FIRMS |
| MODIS (NDVI) | 250 | MODIS/061/MOD13Q1 |
| Sentinel-2 | 10 | COPERNICUS/S2_SR_HARMONIZED |
| Landsat-8 Landsat-9 | 30 | LANDSAT/LC08/C02/T1_TOA LANDSAT/LC09/C02/T1_TOA |
| NOAA (Humidity) | 27,830 | NOAA/GFS0P25 |
| Precipitation | 11,132 | ECMWF/ERA5_LAND/HOURLY |

Details about each of the datasets are described in the following sections.

### 4.1. MODIS Data

#### 4.1.1. FIRMS

This study utilizes the MODIS FIRMS [31] as the primary data source for identifying hot spots of fire activity. FIRMS was established in 2007 by the University of Maryland with financial support from NASA's Applied Sciences Program and the United Nations Food and Agriculture Organization (UN FAO) to offer timely and accurate active fire location information to natural resource managers who faced difficulties in obtaining satellite-derived fire data. The probability of fire ranges from 0 to 100 percent and the spatial resolution of the product is 1000 m.

Contextual algorithms used by FIRMS [53] compare the temperature of an area suspected of having a fire with the temperature of the surrounding land cover. When the temperature difference exceeds a predetermined threshold, the suspected area is confirmed as an active fire or a "hot spot". Figure 2 shows the distribution of fires determined according to FIRMS data for summer 2021 (1 June 2021–30 August 2021) and for July 2022 (1 July 2022–31 July 2022).

As can be seen from the geographical location of the identified hot spots, in 2021, a higher number of fires occurred especially in the southern region of Ukraine, but the fires were evenly distributed throughout the region. In 2022, they were located and most concentrated along the lines of hostilities. That is why additional research into the possibility of determining the cause of the fire was included in this work, in particular whether the weather conditions contributed to their occurrence.

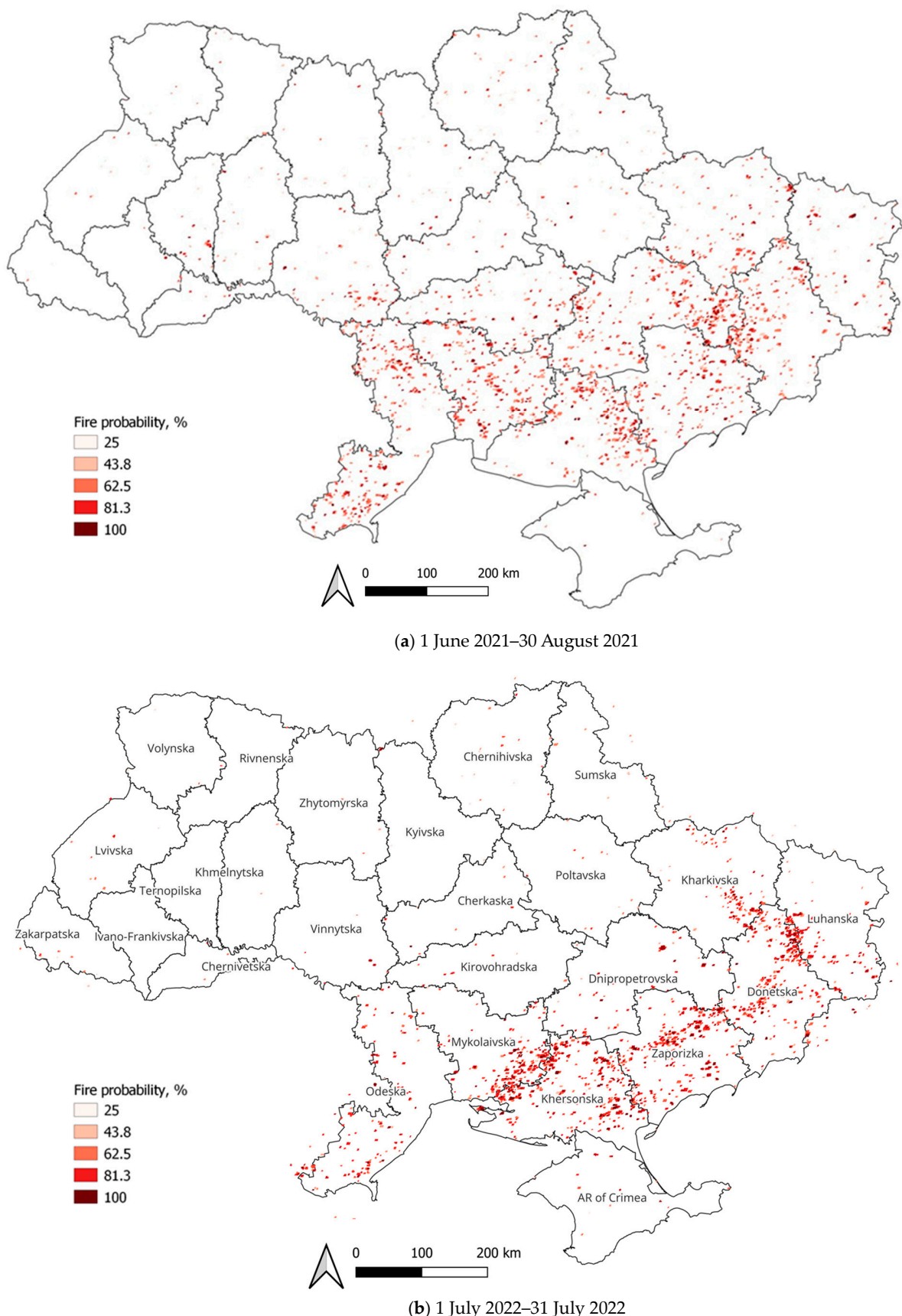

**Figure 2.** Hotspots for Ukraine based on MODIS FIRMS data for (**a**) 2021, (**b**) 2022 for July.

#### 4.1.2. Normalized Difference Vegetation Index

The Normalized Difference Vegetation Index (NDVI) [54] which refers to National Oceanic and Atmospheric Administration—Advanced Very-High-Resolution Radiometer (NOAA-AVHRR) is used to analyze the dynamics of surface greenness. The NDVI method is a metric methodology that measures the balance of energy received and released by the observable objects on Earth. This index sets a value for how green area is when applied to plant communities, that is, the quantity of vegetation present in a particular area and its condition of development. NDVI methods have been used to offer a comprehensive view of land cover change. It is calculated based on near-infrared (NIR) and red (RED) light reflectance assessments as in (1),

$$NDVI = \frac{NIR - RED}{NIR + RED},$$

(1)

where NIR and RED are the amounts of light reflected by growing vegetation and captured by the satellite sensor [55].

### 4.2. Satellite Data

#### 4.2.1. Sentinel-2

The Multispectral Instrument (MSI) aboard Sentinel-2 satellites samples 13 spectral bands: four bands at a 10 m, six bands at a 20 m and three bands at a 60 m spatial resolution (Table 2).

**Table 2.** Sentinel-2 spectral band characteristics.

| Band | Resolution | Central Wavelength | Band Name |
| --- | --- | --- | --- |
| B1 | 60 m | 443 nm | Coastal aerosol |
| B2 | 10 m | 490 nm | Blue |
| B3 | 10 m | 560 nm | Green |
| B4 | 10 m | 665 nm | Red |
| B5 | 20 m | 705 nm | Red edge 1 |
| B6 | 20 m | 740 nm | Red edge 2 |
| B7 | 20 m | 783 nm | Red edge 3 |
| B8 | 10 m | 842 nm | NIR |
| B8a | 20 m | 865 nm | Near-infrared narrow |
| B9 | 60 m | 940 nm | Water vapor |
| B10 | 60 m | 1375 nm | SWIR cirrus |
| B11 | 20 m | 1610 nm | SWIR 1 |
| B12 | 20 m | 2190 nm | SWIR 2 |

Optically corrected Sentinel-2 Level 2A satellite data, featuring a revisit period of 5 days, is employed to identify fire locations and burned areas [55]. Analyzing the change in the Normalized Burn Ratio (NBR) index, which is based on the B8 Visible and Near-Infrared (VNIR 842 nm) band and the B12 Short-Wave Infrared (SWIR 2190 nm) band, we determine the burnt areas (described in Section 5.2). Along with a Level-2A surface reflectance product, Sen2Cor offers an 11-class Scene Classification (SCL) map [56]. Sen2Cor is a Level-2A processor the main purpose of which is to correct single-date Sentinel-2 Level-1C Top-Of-Atmosphere (TOA) products from the effects of the atmosphere in order to deliver a Level-2A Bottom-Of-Atmosphere (BOA) reflectance product. Additional outputs are an Aerosol Optical Thickness (AOT) map, a Water Vapour (WV) map and a Scene Classification (SCL) map with Quality Indicators for cloud and snow probabilities. The data presented in this map hold significant value for automated processing methodologies.

It served as a cloud and shadow mask, enabling the extraction of clear land pixels from Sentinel-2 imagery.

### 4.2.2. Landsat

The Landsat 8 satellite payload consists of two science instruments—the Operational Land Imager (OLI) and the Thermal Infrared Sensor (TIRS). These two sensors provide seasonal coverage of the global landmass at a spatial resolution of 30 m (visible, NIR, SWIR); 100 m (thermal); and 15 m (panchromatic). While OLI provides data at a 30 m spatial resolution, the TIRS band 10 (10.6–11.19 mm) was resampled at 30 m and was used to validate the created burn area mask [57]. Since the combined revisit time of the Landsat-8 and -9 together is 8 days, which is almost two times less often than Sentinel-2, the validation was carried out selectively for cloud-free images and for the test part of the territory.

### 4.2.3. Humidity Data

The National Oceanic and Atmospheric Administration (NOAA) provides humidity data with a spatial resolution that varies based on the source and method of collection [58]. The 384 h forecasts, with 1 h (up to 120 h) and 3 h (after 120 h) forecast intervals, are made at a 6 h temporal resolution and a 27,830 m spatial resolution.

### 4.2.4. Precipitation

In order to exclude possible lightning strikes, which are a meteorological factor for the occurrence of fires in ecosystems, we consider the accumulated precipitation from ERA5-Land data [59] with spatial resolution of 11 km.

### 4.3. Land Cover and Fuel Type Maps

To identify the type of land cover affected by fires, a land cover and crop type map (Figure 3) is used, which was created by using cloud technologies and neural network models [60–62]. For classification processing, 2 bands (VV, VH) of SAR Sentinel-1 descending data with 10 m spatial resolution are used with preliminary preprocessing steps using SNAP 9.0.0 software: Correction of coordinates in the orbit, Specl-filtration, Radiometric calibration, the Range-Doppler Terrain Correction, Data transfer in decibel, and Creating a Data Stack. Also, for classification processing 4 bands (Red B4, Green B3, Blue B2, InfraRed B8) of Sentinel-2 satellites with preprocessing Level-2A and 10 m spatial resolution are used. The revisit time of Sentinel-2 is 5 days, but due to high cloud cover, monthly composites were obtained as the median value of all possible values for every 5 days in the respective bands. A Scene Classification Map (SCL) band with a spatial resolution of 20 m was used to mask clouds from optical data. Optical composites were obtained in the Google Earth Engine cloud platform along the extent of each path of Sentinel-1 data.

Multilayer perceptron (MLP) was used for training the neural network. Compared to deep neural network algorithms, convolutional neural networks in particular, the MLP algorithm loses in accuracy by 1%, but requires much less powerful computing resources and time to obtain the final product. That is why we decided to use the MLP neural network algorithm.

Ground truth data collected along roads were used as training data for the neural network. The overall accuracy of the classification map was calculated based on the confusion matrix [63]. In turn, the confusion matrix was created by comparing the obtained classification map with the collected independent test ground truth data. The overall accuracy of land cover classification maps was 95%.

The given classification maps are adapted to the map of fuel types established by the NFDRS (National Fire Danger Rating System).

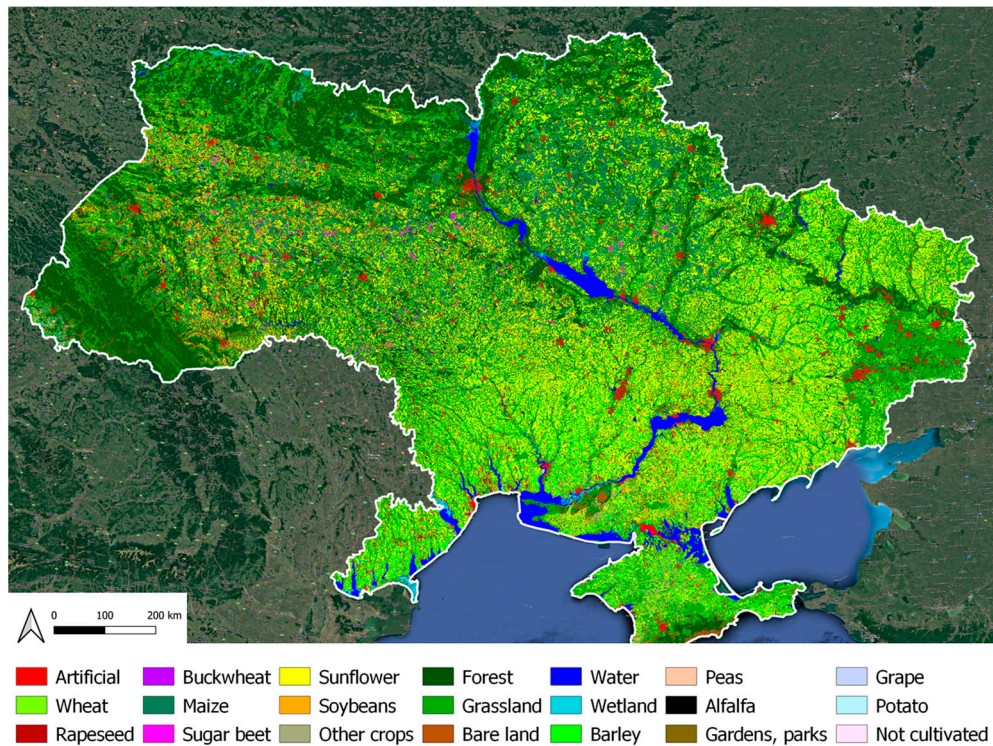

**Figure 3.** Land cover and crop type classification maps for 2021.

*4.4. Fire Potential Index (FPI)*

In this work, the FPI index is used to assess the fire danger risk level, which was calculated for the first time for the full territory of Ukraine according to the methodology used in the American NFDRS information system [64]. Thus, initially, the inputs to the FPI model are a 1 km resolution fuel model map, a Relative Greenness (RG) map [65] that indicates current vegetation greenness compared to historical maximum and minimum values, and a 10 h time lag dead fuel moisture map [66]. The output is a national-scale, l km resolution map that presents FPI values ranging from 1 to 100 [28]. In work [67], the authors used MODIS data with a spatial resolution of 500 m for the territory of Spain.

In our work, we suggest using data with a higher spatial resolution (MODIS NDVI data, 250 m), as well as using an adapted fuel map based on our own land cover classification map with a spatial resolution of 10 m [60–62]. The following data are used for its calculation for the territory of Ukraine:

- Relative humidity 2 m above ground, which has a cross-section of 27,830 m from the NOAA;
- Normalized Difference Vegetation Index (NDVI) based on MODIS data with a spatial resolution of 250 m:
  - Minimum value for last 5 years;
  - Maximum value for last 5 years;
  - Maximum value for current time (last 7 days);
- Land cover classification map for identification of fuel moisture parameters.

The main advantage of the FPI index is its ease of implementation. In particular, the GEE cloud platform [68], which is very user-friendly, contains all the necessary components for calculating the FPI index. Also, since this index was used in the state system of America, we can assume that it is reliable and can be used for fire danger prediction.

*4.5. Datasets for Additional Validation*

4.5.1. ACLED Database

The Armed Conflict Locations and Events Data (ACLED) military operations [69] database was chosen as an additional source of information on human activity that can cause fire. The database provides the geolocation of the settlements in which or not far from which the military actions took place. This database collects a variety of events, including Armed clash, Looting/property destruction, Remote explosive/landmine, but we only selected Shelling/artillery/missile attack, which primarily affects the occurrence of fires. The Figure 4 shows Shelling/artillery/missile attacks for July 2022. As we can see, the concentration of the given points is the greatest precisely in the military zone, and also corresponds to fires during these same periods (Figure 2).

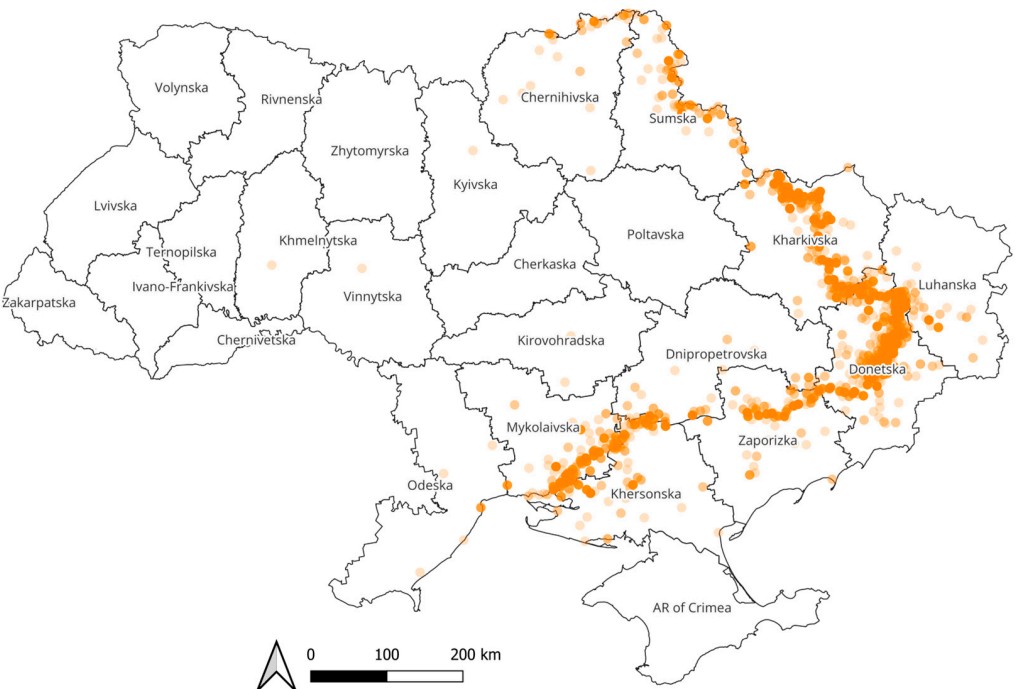

**Figure 4.** Shelling/artillery/missile attack in Ukraine in July 2022 according to the ACLED database (orange points).

4.5.2. Vector Data with Boundaries of Fires from Shelling

In order to perform additional validation of the burned areas map, within this work, the provided vector data with fires created manually on the basis of high-resolution satellite data (Planet/WorldView-3) and Sentinel-2 data was used. This dataset represents fires from shelling in the south (Khersonska oblast) and east (Kharkivska and Donetska oblasts) of Ukraine in 2022 (Figure 5). In particular, for the south of Ukraine, we used fire boundaries for 19 May 2022 (1029 polygons), and for the east we employed the data for the period from 5 July 2022 to 30 July 2022 (984 polygons). These data were collected by our partners within the "Satellites for Wilderness Inspection and Forest Threat Tracking" (SWIFTT) project.

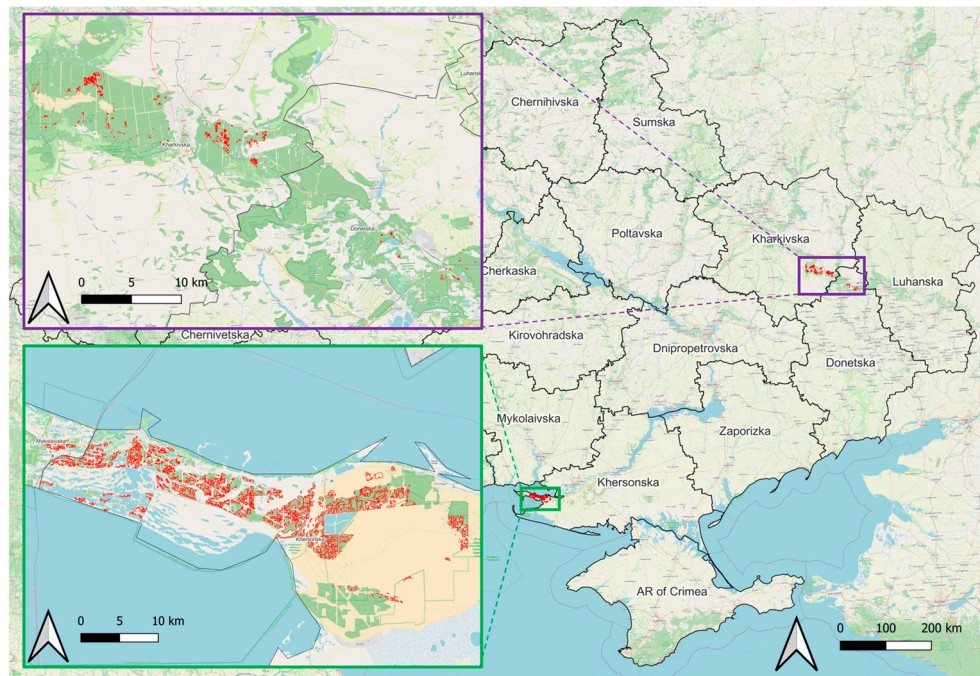

**Figure 5.** Vector data with forest fires from shelling in Ukraine.

## 5. Implementation of the Framework in GEE

The main goal of this work was to create a framework in the Google Earth Engine cloud platform for the territory of Ukraine, which would combine the detection of real fires with the FPI index and would answer the following question: "Was the fire a consequence of meteorological conditions?".

The general block diagram of the proposed framework is shown in Figure 6 and consists of a total of three major blocks—forecasting of the fire danger level (green block), identifying the fire locations (blue block), and decision making of the nature of the fire occurrence (orange block). We consider each of these blocks in more detail in the corresponding sections below.

### 5.1. Fire Potential Index

This section describes the steps (Figure 6, green block) in the GEE cloud platform for calculating the FPI for the territory of Ukraine based on satellite data.

The FPI model requires the knowledge of three vegetation variables: the live ratio, the moisture content of small dead vegetation, and the fuel type [70]. The live ratio is computed by comparing the current and maximum values of the Normalized Difference Vegetation Index of an area in a given period. The moisture content of small dead fuels is estimated from meteorological parameters. The fuel type is based on Land cover/Crop type classification. The resulting FPI values range from 1 to 100 [28], which, according to work [70], can be considered a probability of fire.

The Fire Potential Index (FPI) model is utilized to identify regions with the highest likelihood of fires. The formula for calculating the FPI index takes as input NDVI MODIS data (5-year maximum $ND_{max}$, 5-year minimum $ND_{min}$, 7-day maximum $ND_0$ NDVI values for each pixel) and mean currently fuel moisture information $FM_{10}$ based on the last 10 h according to NOAA humidity data and fuel map information $MX_d$ (table data [66] based on fuel map). The general formula employed for FPI calculation is as follows [12,71]:

$$FPI = (1 - TN_f) \cdot (1 - LR) \cdot 100, \qquad (2)$$

where $TN_f$ is calculated according to formula

$$TN_f = \frac{FM_{10} - 2}{MX_d - 2},$$ (3)

and live ratio (*LR*) determines the ratio of living vegetation to dead vegetation using the following formula:

$$LR = RG_f \cdot LR_{max}/100,$$ (4)

where $LR_{max}$ is (according to [14])

$$LR_{max} = 35 + 40\frac{ND_{max} - 100}{80}.$$ (5)

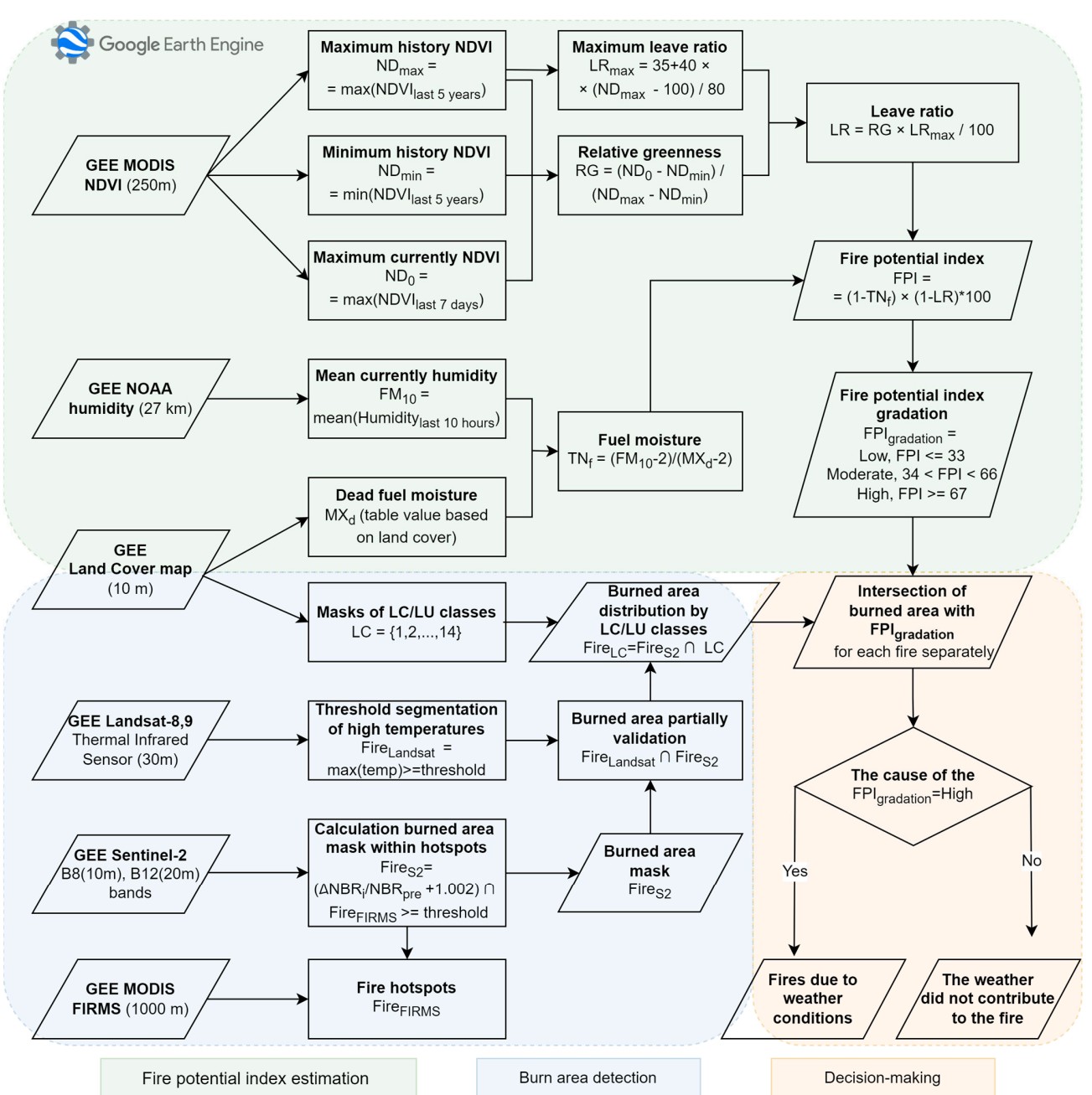

**Figure 6.** The framework for analysis of fire cause based on satellite data in Google Earth Engine cloud platform.

The following formula of live ratio is utilized to determine Relative Greenness ($RG_f$), which is then converted into a fractional value, $RG_f = RG/100$, in order to obtain an estimate of the proportion of green vegetation in a given area. $RG$ is calculated using NDVI indices derived from MODIS data:

$$RG = (ND_0 - ND_{\min})/(ND_{max} - ND_{min}) \cdot 100. \tag{6}$$

As a result, the values of the FPI index ranging from 0 to 100 are obtained. Therefore, we divide the Fire Potential Index maps within each region into three gradations to make inferences about causes of fire easier and faster than dealing with a continuous measure and for easier decision making. For division into gradations, a uniform division of the range of the value, which is divided into gradations, is used [72,73]. The first includes values from 0 to 33 (Low), the second from 34 to 66 (Moderate) and the third from 67 and above (High):

$$FPI_{gradations} = \begin{bmatrix} Low, & 0 < FPI \leq 33 \\ Moderate, & 34 < FPI < 66 \\ High, & FPI \geq 67 \end{bmatrix}. \tag{7}$$

*5.2. Burned Area Detection*

The use of FIRMS data makes it possible to determine the potential areas for which fires are possible. In our case, we used all pixels (that is, with probability between 0 and 100), which made it possible to create a fire hotspot mask $Fire_{FIRMS}$ for the territory of Ukraine. Given the need to operationally and quickly obtain accurate data on the location of the fire, deep neural network approaches are not used in this work, as they require training data (which are not available for the entire country), powerful computing resources and access to professional cloud platforms such as Amazon Web Services or CREODIAS. Calculation of fires based on high spatial resolution data is a resource-intensive process; therefore, in order to reduce the volume of satellite data processing, we suggest searching for fires and burned areas within hotspots obtained from FIRMS data.

In order to enhance the precision of fire location and burn area determination, we utilized products obtained from Sentinel-2 satellite data. Specifically, we relied on the Normalized Burn Ratio (NBR) index [74] as the foundation for our approach, because it is simple and quick to implement and enables quick and efficient results. Initially, we calculated NBR indices for the region affected by the fire ($NBR_{pre}$—before fire, $NBR_{post_i}$—during or after fire):

$$NBR_{pre} = \frac{B8 - B12}{B8 - B12}, \; NBR_{post_i} = \frac{B8_i - B12_i}{B8_i - B12_i}, \tag{8}$$

where B8—Visible and Near-Infrared (VNIR 842 nm) band and B12—Short-Wave Infrared (SWIR 2190 nm) Sentinel-2 MultiSpectral Instrument (MSI) bands for each image, specifically the *i*th one; fire zones are to be detected.

Subsequently, the assessment of land surface changes was conducted individually for each *i*th image:

$$\Delta NBR_i = NBR_{pre} - NBR_{post_i}. \tag{9}$$

The output of this process is a raster mask with multiple grades, wherein high values represent the active fire zones and low values indicate the area where the fire has occurred. Fire detection was conducted based on the ratio for each individual *i*th image in GEE [75] (adding 1.001 to the denominator ensures that the denominator will never be zero, thereby preventing the equation from reaching infinity and failing [76]):

$$Fire_i = \frac{\Delta NBR_i}{NBR_{pre} + 1.001} \geq threshold. \tag{10}$$

The general map of fires is calculated according to the following formula (combination of all fires for all used satellite data from 1 to n) [77]:

$$Fire_{S2} = \sum_{i=1}^{n} Fire_i. \tag{11}$$

In order to validate the received fire mask, it is suggested to use TIRS in band 10 Landsat-8 and -9 data (resampled from 100 m to 30 m). To determine the fire, the maximum temperature for the studied area is determined and the temperature with low values is filtered.

This can be represented in the following formula:

$$Fire_{Landsat} = max(temp) \geq threshold, \tag{12}$$

where $max(temp)$ is the maximum temperature value determined according to Landsat 8/9 TIRS data, and $threshold$ is the value chosen by the user depending on the season (in summer, this value increases, and vice versa in spring and autumn) [71]. We can set fixed values for all maps, but we are interested in obtaining the highest-quality product possible. Therefore, different threshold values were chosen to prevent overestimation or high underestimation of burnt areas. The threshold value is selected by analyzing extremely high values, for example, by the three-sigma algorithm [78], or Otsu's method [79]. After that, the validation of the fires obtained from the Sentinel-2 data is carried out, taking into account the limited available data due to cloud cover and the low regularity of Landsat satellite [78]. Therefore, we proposed the following formula for partial verification of the fires detected by Sentinel-2 (the intersection of features from the Sentinel-2 and Landsat data):

$$Fire_{verified} = Fire_{S2} \cap Fire_{Landsat} \neq \varnothing. \tag{13}$$

The ideal case is when all satellites provide data for the same time period. However, taking into account the specifics of satellite monitoring, it is worth taking the studied interval of at least 5 days, which corresponds to the revisit time of Sentinel-2. By additionally using the land cover classification map (LC), we obtained the opportunity to understand exactly what type of land cover was affected by the fire according formula (intersection of burned area with land cover map):

$$Fire_{LC} = Fire_{S2} \cap LC. \tag{14}$$

### 5.3. Decision Making about the Fires Nature

As a result of the first two blocks, gradations of the FPI index were calculated and active fires or burned areas were determined on the territory of Ukraine. Intersection of the burned area with $FPI_{gradation}$ for each fire separately is calculated in this block. When the specified fire fell into a zone with a high gradation of the FPI index, we believe that the fire could really have occurred as a result of weather conditions. Otherwise, we assumed that the weather did not contribute to the occurrence and spread of the fire. In such a case, in particular, in the case of active military actions on the territory of Ukraine, the next step was the analysis of open sources of information regarding the events that could have occurred in such territories and caused the fire. An example of such data is the ACLED database.

### 6. Results

The summer months of 2021 and 2022 were chosen for the study of fires, since it is during these periods that meteorological conditions (drought conditions and lightning strikes) favor the spontaneous occurrence of fires in Ukraine. In particular, for the eastern and southern regions of Ukraine, the areas of burned areas and determined land cover types were calculated for the period of 15–17 July 2022. It was during this period that the most active fires occurred. The developed methodology was also tested on the territory

near the National nature park of nationwide significance "Oleshkivski pisky", and the Biosphere reserve of nationwide significance "Chornomorskyi", Kherson region, which burned down in the first half of August 2022.

*6.1. Burned Area Based on Sentinel-2 Data*

6.1.1. Burned Area Identification

To identify burned area, we used the Normalized Burn Ratio (NBR) index [74] in the GEE Cloud platform [75]. After calculating the corresponding areas in 15–17 July 2022 that were burned according to Sentinel-2 data, the following distribution by land cover types was obtained (Table 3). Land cover classes and crop types were extracted from a classification map with a spatial resolution of 10 m, obtained in 2022 (as of July). The overall burned area was estimated to be 104.2 thousand hectares. This includes approximately 70 thousand hectares of cereal crops, 0.26 thousand hectares of rapeseed, 7.8 thousand hectares of summer crops, 25 thousand hectares of grassland, and 1.1 thousand hectares of forest.

**Table 3.** Burned area according to Sentinel-2 as of 15–17 July 2022.

| | Cereal | Rapeseed | Summer Crops | Non Cultivated | Grassland | Forest | Total |
|---|---|---|---|---|---|---|---|
| Donetska | 25,823.7 | 4.3 | 144.8 | 864.07 | 9258.51 | 266.3 | 36,361.6 |
| Khersonska | 10,843.2 | 87.3 | 4129.4 | 3831.48 | 853.43 | 150.6 | 19,895.3 |
| Mykolaivska | 14,556.3 | 41.0 | 1398.2 | 2382.65 | 582.25 | 286.8 | 19,247.1 |
| Zaporizka | 9173.0 | 81.2 | 1774.7 | 810.91 | 637.55 | 61.4 | 12,538.8 |
| Luhanska | 3396.0 | 30.1 | 175.3 | 308.37 | 4550.68 | 287.7 | 8748.2 |
| Kharkivska | 5395.0 | 0.1 | 43.2 | 39.58 | 729.44 | 87.1 | 6294.4 |
| Dnipropetrovska | 745.6 | 7.7 | 152.4 | 53.32 | 109.64 | 2.5 | 1071.2 |
| Poltavska | 45.1 | 13.1 | 1.0 | 0.7 | 0.18 | 0.0 | 60.1 |
| Odeska | 3.0 | 0.0 | 7.7 | 0.12 | 1.67 | 0.0 | 12.6 |
| Total | 69,980.9 | 264.8 | 7826.6 | 8291.2 | 16,723.4 | 1142.4 | 104,229.2 |

However, the Sentinel-2 estimates represent a lower estimate only, as cloud cover during satellite surveys may have affected the accuracy of the results (Figure 7). Given that the average wheat harvest in 2021 was 45.3 tons per hectare (according to the State Statistical Service of Ukraine), and that the fires occurred during the wheat and barley harvest period, the potential losses due to fires along the front line were estimated to be 3.17 million quintals or 317 thousand tons of grain. Examples of determining burned areas for agricultural fields are shown in Figure 8.

Also, the proposed approach was tested for the territory of the ecosystem near the National nature park of nationwide significance "Oleshkivski pisky", Kherson region (Figure 1) for the first half of August 2022. The total burned area of forest is 614 ha, Grassland: 784 ha, and wetland: 33 ha. Also, the part of found burned areas was confirmed by the increased temperature of the earth's surface according to Landsat-9 (Figure 9). Taking into account the difference in the time of providing Landsat 8/9 and Sentinel-2 data, we cannot expect a complete correspondence of the increased temperature to the burned areas, but only the part that is closer in time to the time of the Landsat survey.

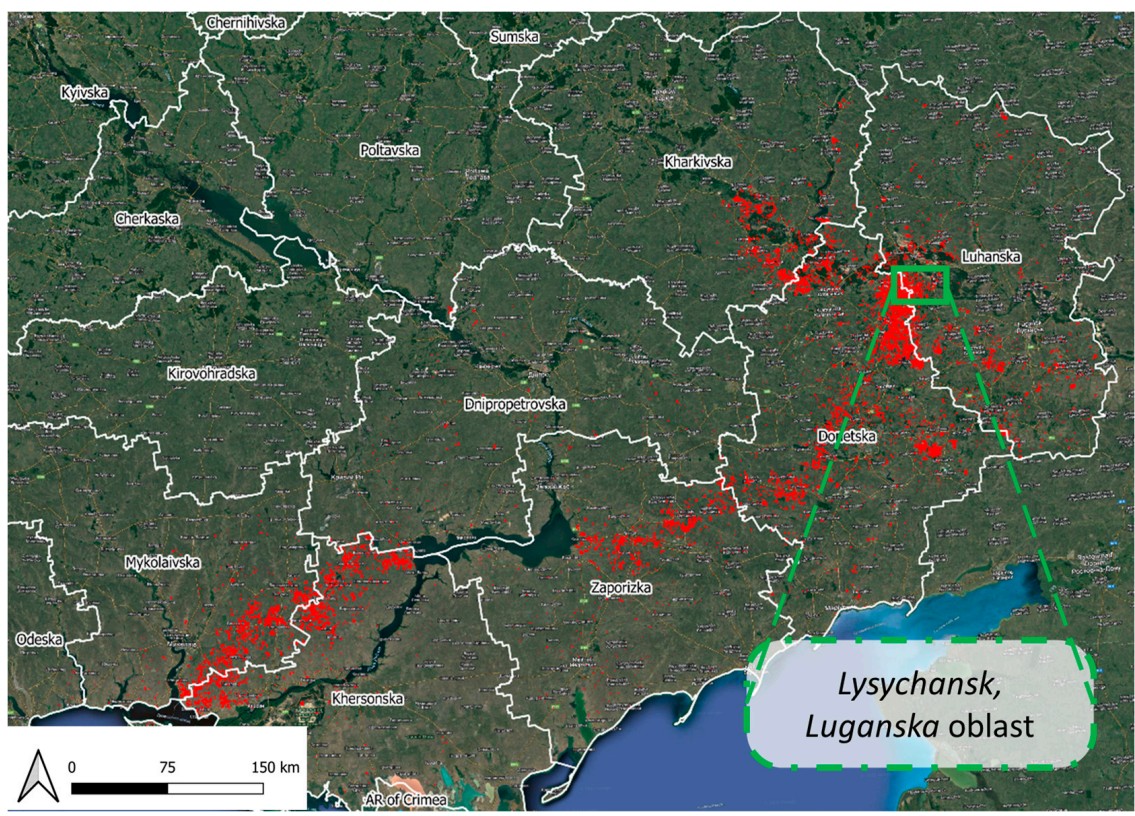

**Figure 7.** Burned mask (in red) retrieved from Sentinel-2 (15–17 July 2022).

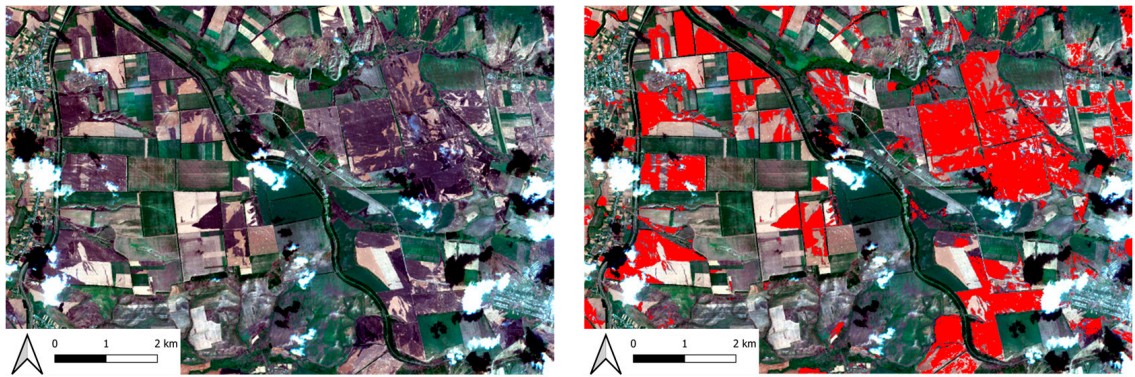

**Figure 8.** Burned mask (in red) retrieved from Sentinel-2, near Lysychansk, Luganska oblast. Sentinel-2 RGB for 17 July 2022 used as background.

The methodology has also been tested on other types of land cover, in particular on grassland. Figure 10 shows a completely burned Biosphere reserve of nationwide significance "Chornomorskyi", Kherson region, where the burned area of grassland is 2153 ha, and the burned area of forest is 157 ha.

### 6.1.2. Validation of Fires Using High-Resolution Data

To validate our approach to creating a fire mask from Sentinel-2 satellite data, we used vector boundaries with forest fires caused by shelling, prepared of our partners within SWIFTT project. In particular, for the south of Ukraine (mainly the Kherson region), the forest was most affected in the month of May. Analyzing satellite data, we recorded the fire itself, as early as 5 September 2022 (Figure 11). However, on this day, it was very difficult to make an assessment because of the smoke in the picture.

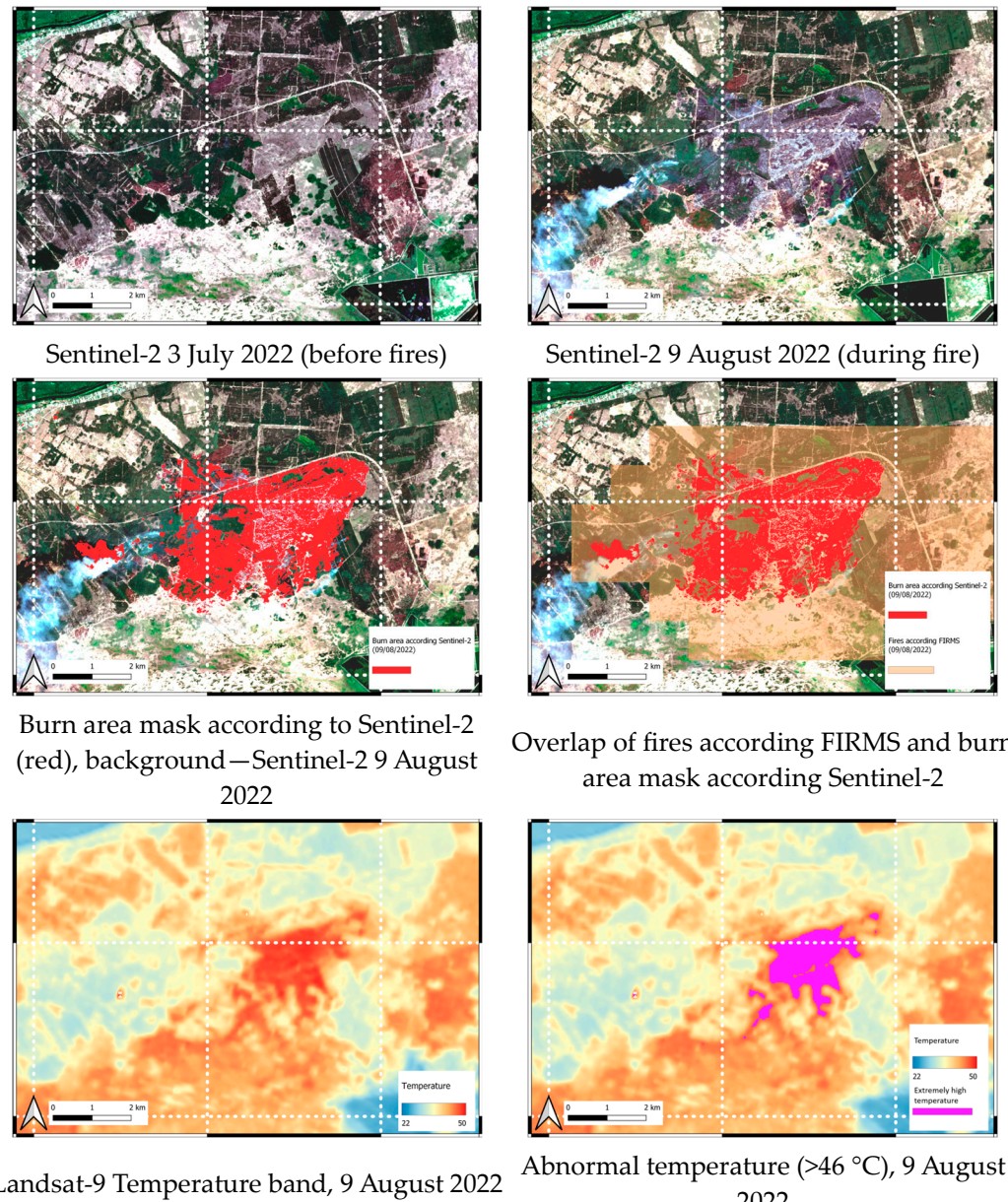

**Figure 9.** Burned area detection according to Sentinel-2, FIRMS and Landsat-9 data near the National nature park of nationwide significance "Oleshkivski pisky", Kherson region.

To create a mask of burned areas, we used Sentinel-2 images (from 24 April 2022 to 19 May 2022). The resulting mask and vector boundaries used for validation are presented in Figure 12.

A similar mask was created for the north of Ukraine. Sentinel-2 data from 2 July 2022 to 30 July 2022 were used only for this purpose.

Using the created masks and vector data with fires, a confusion matrix with two classes Fire and Not Fire was constructed (Table 4), where PA—producer accuracy, UA—user accuracy, F1—F1-score accuracy, OA—overall accuracy [80].

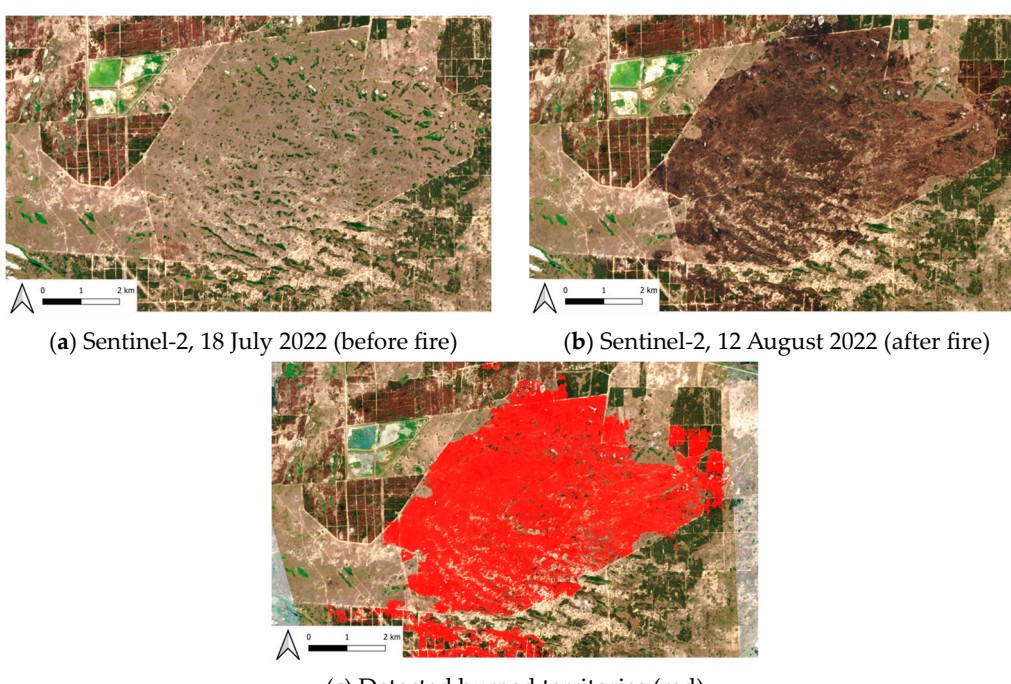

(**a**) Sentinel-2, 18 July 2022 (before fire)    (**b**) Sentinel-2, 12 August 2022 (after fire)

(**c**) Detected burned territories (red)

**Figure 10.** Example of land cover changing based on (**a**) Sentinel-2 before fire and (**b**) Sentinel-2 after fire damages and (**c**) detected burned area in biosphere reserve of nationwide significance "Chornomorskyi", Kherson region.

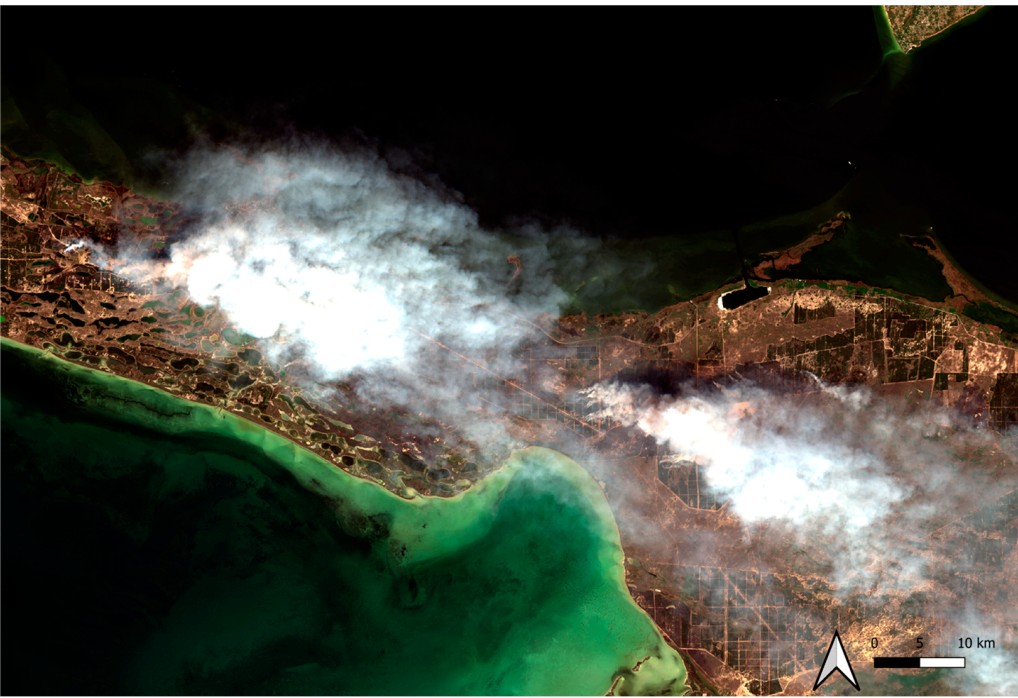

**Figure 11.** RGB Sentinel-2 for 9 May 2022 in southern Ukraine, Kherson region.

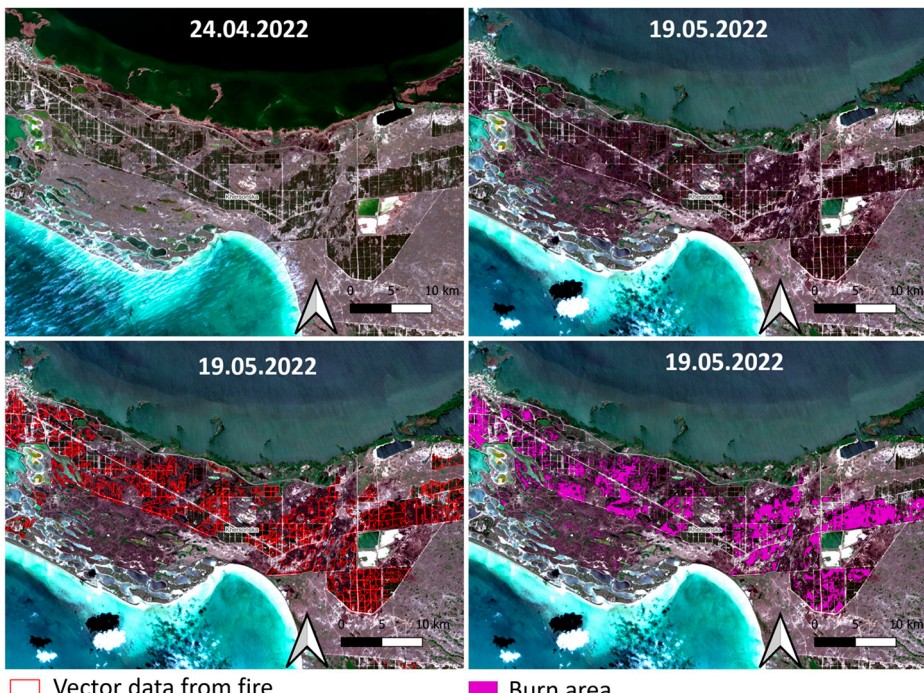

**Figure 12.** Burnt forests in the south of Ukraine, Kherson region.

**Table 4.** Confusion matrix for fire mask estimation.

| CLASS | South | | | East | | |
|---|---|---|---|---|---|---|
| | **PA** | **UA** | **F1** | **PA** | **UA** | **F1** |
| Fire | 53.9 | 71.6 | 61.5 | 75.3 | 34.4 | 47.3 |
| Not Fire | 98.5 | 96.8 | 97.6 | 99 | 99.8 | 99.4 |
| OA | | 95.5 | | | 98.9 | |

The resulting inaccuracies are explained by the fact that data with a higher spatial resolution, in comparison with Sentinel-2, were used to create polygons with fires. We can also see that for the south, the accuracy for the class of fires is higher (F-score = 61.5%), which may be due to the fact that Planet data were used for the south, which have a worse spatial resolution compared to WorldView-3 data, which were used for the east of Ukraine. Although the overall accuracy (OA) in this case was high, it is due to the Not Fire class, which is well demonstrated by PA and UA.

## 6.2. Fire Potential Index for Summer for Ukraine

### 6.2.1. FPI for Ukraine

Using satellite data and a land cover classification map, a Fire Potential Index was created for the territory of Ukraine for 2021 and 2022. For this, the GEE cloud platform was used. The resulting index is a raster map with a spatial resolution of 250 m, containing values from 0 to 100 that indicate the indicator of fire occurrence. Figure 13 shows these maps, where blue values have low values and red values have high values.

Observations from the provided image reveal that during the summer of 2021, Ukraine predominantly experienced fires dispersed across its southern and eastern regions, aligning with the Fire Potential Index.

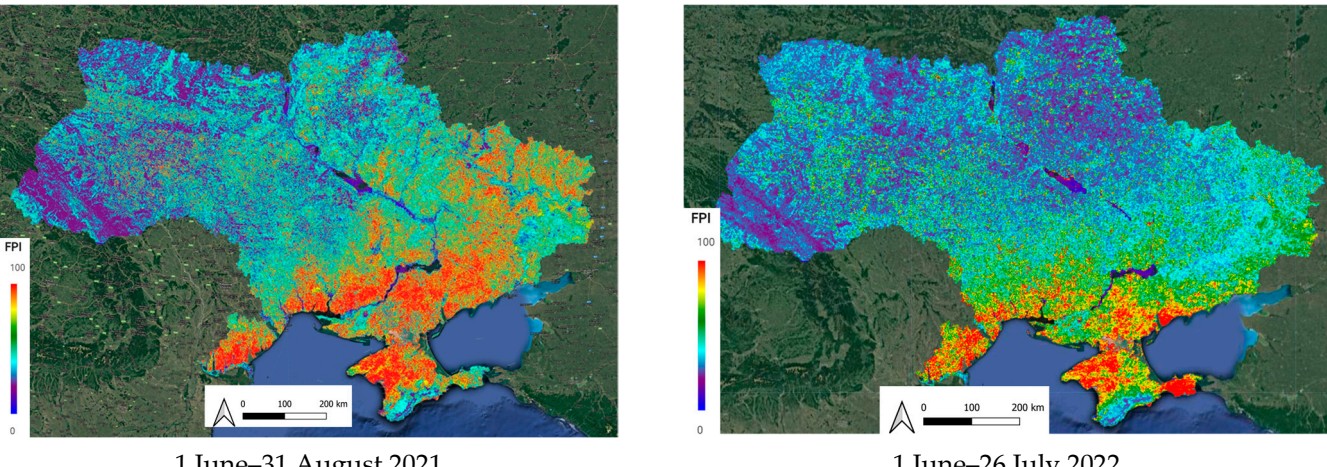

1 June–31 August 2021                                                    1 June–26 July 2022

**Figure 13.** Fire Potential Index for summer 2021 and 2022.

6.2.2. Validation of FPI

To confirm the effectiveness of the FPI for the territory of Ukraine, a comparison of the number of real fires according to satellite data (FIRMS) and the majority value of the FPI in the regions of Ukraine was carried out. For this, correlation was calculated according the following formula:

$$R = \frac{\sum_{i=1}^{n} \left( F_i - \overline{F} \right) \left( N_i - \overline{N} \right)}{\sqrt{\sum_{i=1}^{n} \left( F_i - \overline{F} \right)^2} \sqrt{\sum_{i=1}^{n} \left( N_i - \overline{N} \right)^2}}, \tag{15}$$

where $F_i$—majority value of the FPI for the $i$th oblast, $\overline{F} = \sum_{i=1}^{n} F_i / n$—mean value of FPI majority values, and $N_i$—number of fires according to FIRMS data for the $i$th oblast, $\overline{N} = \sum_{i=1}^{n} N_i / n$—mean value of the number of fires for n = 25 oblasts.

The results for 2021 show a correlation of 0.66, and factors contributing to these fires include meteorological conditions, as anticipated by the Fire Potential Index, and anthropogenic influences such as stubble burning in agricultural fields. For 2022, the correlation is 0.25 for all oblasts. This is a rather low value and at first glance one may get the impression that there is no connection. But if we exclude oblasts in which active hostilities are taking place (Luhanska, Mykolaivska, Zaporizka, Kharkivska, Donetska, Khersonska), the correlation coefficient for 2022 will be 0.68. Therefore, in 2022, the correlation is low due to the eastern and southern oblasts, and in all other territories the trend remains the same as for 2021.

On the other hand, in 2022, noticeable deviations are observed, the Fire Potential Index indicates possible fires related to weather conditions only in the southern districts of Mykolaiv, Kherson and Odesa regions. Actual fires appear to trace a front line, suggesting their origin as a consequence of human activities like rocket and bomb usage (Figure 2) and also supporting this with information from the ACLED database (Figure 4).

6.2.3. FPI Gradation

With maps using a value from 0 to 100, it is difficult to understand the ways in which potential fire is possible for each region and to make some additional calculations. Therefore, the FPI was divided into three gradations, low, moderate, and high fire danger. The obtained results are presented in Figure 14.

Separately selected fires according to FIRMS at the intersection with the FPI index are presented in Figure 15. The figure shows that the high level of fire danger at the intersection with real fires is located in Odesa, Mykolaiv and Kherson oblasts, while all fires that occurred in the east have a mostly average level or even a very low one.

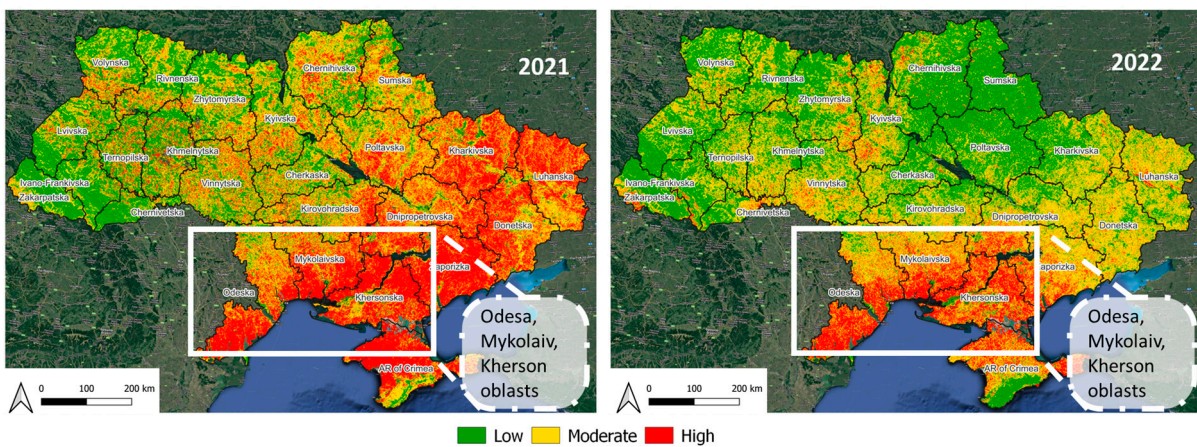

**Figure 14.** Fire Potential Index, summer 2021 and 2022.

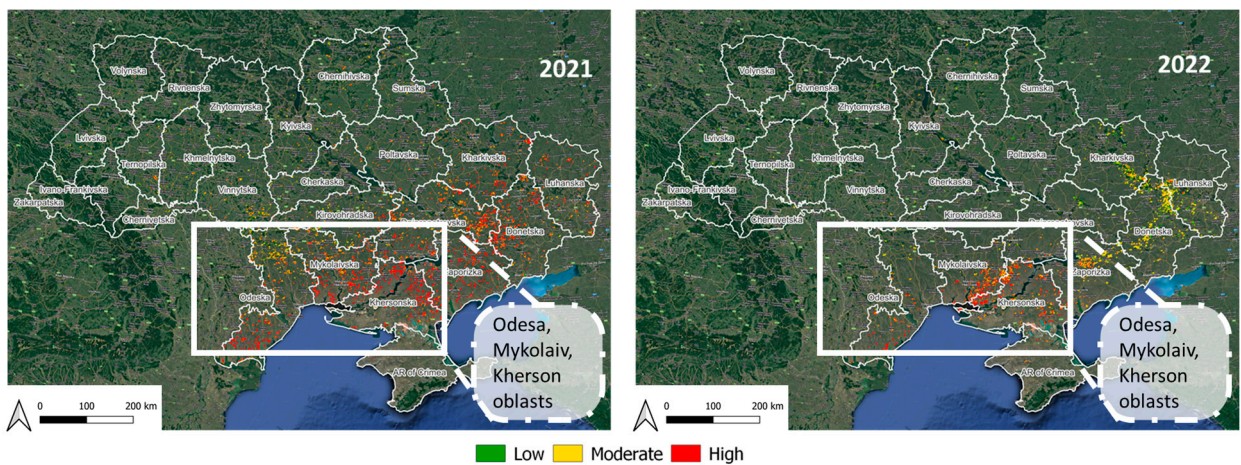

**Figure 15.** Fire Potential Index for real fires for summer 2021 and 2022.

Within each oblast of Ukraine, the percentage of the appropriate gradation of FPI was calculated and compared with the obtained burned areas determined by Sentinel-2. The obtained results are presented in Table 5. It can be seen from the table that in seven oblasts, fires fell into the moderate level of the FPI, one into the high and one into the low. Those fires that fell into a low level of fire danger were considered to be caused by anthropogenic influence, since they have a low influence of weather indicators. Regarding the area with a high level of fire danger, we can assume that the fires could have been caused by weather conditions. For medium fire danger, we cannot definitively state that the fires were due to weather conditions, and additional data need to used such as ACLED to draw conclusions.

If we consider the area of fires in the eastern regions of Ukraine in 2022 obtained from satellite data and the percentage of high probable fires according to the Fire Potential Index maps, we will obtain a correlation between them of 0.1. This is an additional confirmation, not only visually, that the fires in these regions did not arise from natural causes, but that most of the main reasons for such fires are military events in those territories.

**Table 5.** Burned area according to Sentinel-2 as of 15–17 July 2022 and Fire Potential Zone by Fire Potential Index in percentage.

| Region | Fire Potential Zone, % | | | Burned Area, ha |
|---|---|---|---|---|
| | Low, % | Moderate, % | High, % | |
| Donetska | 20.54 | 74.10 | 5.35 | 36,361.6 |
| Khersonska | 7.20 | 37.01 | 55.78 | 19,895.3 |
| Mykolaivska | 5.90 | 55.82 | 38.28 | 19,247.1 |
| Zaporizka | 7.24 | 67.29 | 25.46 | 12,538.8 |
| Luhanska | 23.92 | 69.95 | 6.12 | 8748.2 |
| Kharkivska | 41.90 | 55.80 | 2.29 | 6294.4 |
| Dnipropetrovska | 21.96 | 64.40 | 13.65 | 1071.2 |
| Poltavska | 66.65 | 31.87 | 1.48 | 60.1 |
| Odeska | 10.32 | 45.94 | 43.75 | 12.6 |
| Total | | | | 104,229.2 |

### 6.2.4. Additional Precipitation Data

To exclude the occurrence of fires due to lightning, cumulative precipitation was calculated for the same periods as in the FPI index. As we can see from Figure 16, in 2022, in contrast to 2021, the southern and eastern regions had the least precipitation, which indicates the least possible amount of lightning.

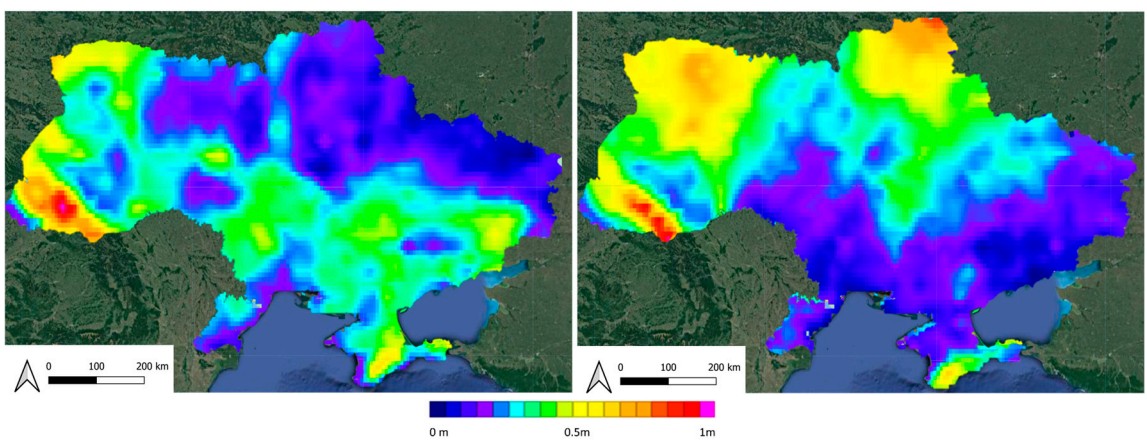

1 June–31 August 2021        1 June–26 July 2022

**Figure 16.** Accumulated Precipitation for summer 2021 and 2022, m.

### 7. Discussion

This article proposes a framework for wildfire monitoring and determining the cause of a fire in the Google Earth Engine cloud platform. The developed information technology combines an optimized approach to monitoring fires and burned areas based on the NBR index with the FPI index, which helps to understand the possible cause of the fire at country level.

As discussed in the introduction, there are many different fire risk monitoring systems already in place at the national level. For Ukraine, such a system has not existed until now, and the authors of this work set the goal of developing a framework that could work in an operational mode. The criteria for choosing the FPI index was, first of all, the ease of its implementation in the GEE cloud platform, since the platform already contains all the necessary data for observations. Of course, for example, the FWI index shows better results,

but its implementation requires specialized cloud platforms (for example, Amazon Web Services or CREODIAS) and resources that are not currently allocated by the government for continuous use.

A similar story is observed with the use of the NBR index to determine burned areas. Of course, with high-quality training data and appropriate computing resources, using modern neural network approaches, it is possible to obtain highly accurate contours of the fires and burned lands themselves. However, if we are talking about a technology that would work systematically at the country level, it is difficult to ensure, and simple, fast-acting, but less accurate approaches in this case have the right to exist.

In 2021, work began on an improved version of the FPI index—the Wildland Fire Potential Index (WFPI) [81], and already in 2023 it began to be actively used. It already accepts meteorological indicators, such as wind, precipitation and temperature. We see this as a way to improve the developed framework.

One of the main problems we faced while performing the work is the insufficient amount of cloud-free optical data. It is for this reason that only a part of the found burned areas was validated in this paper to confirm the correctness of the algorithm for Sentinel-2. Possible further research is to use the Sentinel-1 radar data to solve the cloud issue.

The results presented in this article demonstrate the effectiveness of using satellite data and land cover classification maps to calculate a Fire Potential Index. The Fire Potential Index was created for the years 2021 and 2022 for the summer time for Ukraine, and the resulting maps showed a correlation between the areas with high values of FPI and the actual fire events in Ukraine in 2021, and no correlation for 2022. However, if we separate the territories with military actions, the correlation in 2022 is almost the same as in 2021 (68%). This indicates that the FPI index worked in 2022 at a similar level as in 2021, but of course it does not take into account the anthropogenic impact. Additional evidence of this is also concentrated along the line of fire occurrence and points from the ACLED database.

During the assessment of the accuracy of the mask of burned areas, we realized that the threshold in our methodology remains selection, since its increase rejects pixels with potentially burned areas, but if it is reduced, then noise penetrates into the mask, which can negatively affect the assessment these areas or precisions. During our study, we experimented with various threshold selection methods to identify the most suitable approach for fire detection in our dataset. We evaluated methods such as simple thresholding, adaptive thresholding, and the Otsu's method. Among these methods, the Otsu's method, coupled with our modifications, proved to be the most effective for our fire detection task. However, determining the correct threshold values for the NBR index is not the key task of this work. There is no universal solution to this problem; everything depends on the territory and time period for which it is selected. To refine the mask, it is worth using deep learning methods in the future, at least for those areas most affected by fires, or for areas of increased level of interest.

## 8. Conclusions

This study presented a novel integrated framework for national-scale burned area mapping and fire cause assessment using satellite remote sensing data and fire risk modeling. The methodology combines multi-spectral burned area mapping using the Normalized Burn Ratio index from Sentinel-2 imagery with fire danger analysis based on the Fire Potential Index calculated from vegetation, fuel moisture, and land cover data.

The approach was implemented for all of Ukraine in the Google Earth Engine cloud computing platform to enable efficient large-scale processing. Burned areas totaling 104,229 ha were mapped during 15–17 July 2022, with cereal crops being the most affected land cover type. Comparison with the FPI fire risk map showed fires corresponded well spatially with high-risk areas in 2021 but not in 2022, indicating the 2022 fires were predominantly anthropogenic rather than climate-driven.

The proposed framework demonstrates an effective methodology for national-level fire monitoring by integrating burned area detection with fire risk modeling. Automated

mapping of burned areas with a 10 m resolution enables precise assessment of fire impacts and damages across land cover types. The discrepancy-based technique for identifying anthropogenic fires provides a simple yet powerful approach for determining fire causes using satellite and meteorological data analytics.

Further work should focus on incorporating additional validation data, exploring effects of different burned area and fire risk modeling algorithms, and implementing the methodology for near real-time operational monitoring. The study demonstrates the potential of applying integrated multi-source satellite data analytics in Google Earth Engine to develop scalable and automated systems for assessing wildfire activity and risks globally.

**Author Contributions:** Conceptualization, A.S. and B.Y.; methodology, B.Y., A.S. and L.S.; software, B.Y.; validation, H.Y.; data curation, B.Y. and H.Y.; writing—original draft preparation, B.Y. and L.S.; writing—review and editing, A.S.; visualization, H.Y.; supervision, A.S. All authors have read and agreed to the published version of the manuscript.

**Funding:** This research was funded by Horizon 2020 EuroGEO Showcases: Application Powered by Europe (e-shape) project (grant number 820852), project of the Ministry of Education and Science of Ukraine "Information technologies of geospatial analysis of the development of rural areas and communities" (grant number PH/27-2023).

**Institutional Review Board Statement:** Not applicable.

**Informed Consent Statement:** Not applicable.

**Data Availability Statement:** Not applicable.

**Acknowledgments:** The authors acknowledge the projects H2020 ERA-PLANET www.era-planet.eu (accessed on 18 October 2023), trans-national EU Horizon 2020 project SMURBS (grant number 689443) www.smurbs.eu (accessed on 18 October 2023), HORIZON Europe project "Satellites for Wilderness Inspection and Forest Threat Tracking" (grant number 101082732), National Research Foundation of Ukraine "Geospatial models and information technologies of satellite monitoring of smart city problems" (grant number 2020.02/0284), and "Deep learning methods and models for applied problems of satellite monitoring" (grant number 2020.02/0292).

**Conflicts of Interest:** The authors declare no conflict of interest.

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
