# Peer review of "Google Earth Engine Framework for Satellite Data-Driven Wildfire Monitoring in Ukraine"

_fire, doi:10.3390/fire6110411_

Round 1
Reviewer 1 Report (Previous Reviewer 3)
General comments
I found the paper improved in comparison to the previous version. However, although authors answered to all my comments, there are still many critical issues detailed in the comments below. Moreover, figures and caption need to be significantly improved to facilitate the interpretation of results. Finally, although authors use “Google Earth Engine” to perform their study, this geospatial platform is not sufficiently well described in the text. I hope that the following comments/suggestions will help authors to improve their study.
Specific comments
Line 20: If the proposed methodology is less effective in high fire risk areas, which are of more interest for fire management purposes, this should be mentioned both in the abstract and conclusions.
Line 23: the number of keywords may be reduced.
Line 28: monitoring forest fires…
Line 32: check the characters in bold to see if they fit with the journal style.
Line 44: MODIS should be defined here.
Line 47: with a spatial resolution of 1 km
Line 57: this sentence should be revised.
Line 75: does not work…
Line 78: see comment at line 44;
Line 80: specify the VIIRS spatial resolution for I and M-bands.
Line 96: … at higher spatial resolution are used for a more accurate location and mapping of active fires.
Line: 98: there is no mention of literature studies using Landsat 8 OLI and Sentinel 2 MSI data to map active fires.
Line 106: Sentinel 2 data enable the identification of small fires; however, the low temporal resolution of those data is less suited for a prompt identification of forest fires. Several studies have demonstrated the capacity of SEVIRI in detecting fires at the early stage. However, there is no mention of those studies in the manuscript.
Line 177: see previous comment.
Line 188-214: the introduction is too long. Part of this content (FPI) should be included in section 4.
Line 224: remove for all Ukraine.
Line 240: this sentence requires a reference.
Line 242: various types of earth current. What do you mean here?
Line 267: please, remove this sentence.
Line 268: Contextual algorithms used by FIRMS to detect fires, together should be described in more detail (e.g., used bands, performance, limitations) by the authors.
Line 279: it is very difficult to distinguish the hotspots with a different confidence level in Figure 2. A larger map could be shown here as an example of hotspot detection performed by FIRMS in Ukraine.
Line 281: the NDVI is poorly discussed.
Line 289: Sen2Cor is not described here.
Line 287: there is mention of the MSI sensor aboard Sentinel-2 satellite and of relative features (number of spectral bands; bands used to detect and map fires, etc.).
Line 294: as for previous comment, in reference to OLI and TIRS instruments for Landsat 8/9.
Line 295: define the band 10
Line 333: differences in land cover recorded between 2021 and 2022 are not discussed in the text.
Line 359: authors should provide more information about the Google Earth Engine platform and relative applications, with relative references.
Line 373: If there are no significant differences here, a single map including also the fire locations could be sufficient to show the areas affected by Shelling/artillery/missile attack.
Line 382: were collected..
Line 411: this formula is not clear.
Line 440: see comment at line 287.
Line 446: as for comment at line 411.
Line 453: Authors state that the threshold is selected by the users based on the season. However, this assertion is too generic and not other information is reported the text. In addition, the used formulation is probably subjected to false and/or missed detections, due to the use of a fixed threshold applied to TIR data, but also this aspect is not addressed by the authors. Moroever, it is not clear if the max temperature refers to the max brightness temperature value measured by the TIRS band 10, and Figure 9 does not help readers in interpreting those data.
Line 462: here authors assess the occurrence of fires through the intersection of a burned area product at 20 m spatial resolution with a fire detection product resampled at 30 m spatial resolution. The temporal buffer selected by the authors to compare those products generated analysing Sentinel-2 and Landsat 8/9 data is not specified and discussed in the text.
Line 462: ..detected by Sentinel-2
Line 465: Why the Fire verified is not used in this formula?
Line 497: Why burned areas in Table 2 are highlighted in red?
Line 508: Lysychansk, Luganska oblast are not marked on the maps.
Line 513: the Landsat product is very difficult to interpret. Map should also include a graticule to enable the comparison with the burned areas from S2 data. Moreover, in the figure caption there is no information about the spectral bands used to generate the imagery displayed in background.
Line 522: Figure 10 caption is not clear to me.
Line 560: Maps in Figure 13 should include the legend of the FPI values. In general, all the maps should include also a scale bar and the north arrow.
Line 594: Figure 14, see previous comment.
Line 597: areas indicated by the authors in text are difficult to discriminate over the maps.
Line 600: Figure 15, see previous comment.
Line 625: Figure 16, see previous comment.
Line 655: “take pictures” should be removed from this sentence.
Line 675: the Otsu’s method is not described here and there is no reference about this method in the text.
Author Response
Dear reviewer,
Thank you very much for your in-depth analysis of our work. Following your constructive comments and suggestions which helped to improve our study, we have made our best to address every point you raised. After taking careful attention to your comments and making a great improvement, we updated our manuscript and we expect you can consider our manuscript for publication.

Reviewer 2 Report (New Reviewer)
General comments
Firstly, I would like to express my respect to the authors for working on a topic that is not only technically challenging but must also be emotionally challenging as well.
I found the paper relatively easy to follow but I do have two questions. Firstly, the authors calculate values of FPI on a bounded continuous scale (0-100) but then reduce them to a three-class categorical measure: low, moderate, high. In the text it is stated that it is "...difficult to understand how potentially fire is possible for each region and to make some additional calculations". Why is it so difficult? The three equal-width classes seem arbitrary and I wondered how the results would differ with alternative class boundaries. Working with a continuous measure would avoid such problems. If there were other reasons to prefer a categorical measure perhaps a few words of explanation could be added.
My second question is more general. The paper is written very much in the context of the current military conflict in Ukraine, which I took to indicate that the authors' main motivation was to highlight the damaging effects of additional fire resulting directly from that conflict. However, the paper also mentions the work of the Armed Conflict Locations and Events Data project (ACLED) and the Satellites for Wilderness Inspection and Forest Threat Tracking project (SWIFTT) which are providing spatial data on fires. I was unclear about how the authors' work relates to that of these other projects. Is it in providing an estimate of the additional burnt area due to military conflict based on identifying events that were unlikely to be weather driven? Does this provide more convincing or comprehensive evidence than simply overlaying armed conflict locations with fire polygons?
Specific comments
Line 57: ECMWF
Please spell out European Centre for Medium range Weather Forecasts.
Lines 180-183: To address the second research question, we analyze fire risks and meteorological conditions and determine the areas where fires were not caused by weather. To solve this issue, we use the Fire Potential Index (FPI) [15], which makes it possible to predict the territories when fires can occur.
This sounds very deterministic. Perhaps it would be better to say "where fires were unlikely to be due to weather" and "predict the territories where fires have a reasonable probability of occurrence".
Line 239: Ukraine's forest fund...
Should this be forest estate?
Line 247: various types of earth currents...
What does this mean?
Line 331: The overall accuracy of land cover classification maps is 95%
How was this accuracy statistic calculated?
Lines 419-422: As a result, the values of the FPI index with values from 0 to 100 were obtained. Therefore, we divided the Fire Potential Index maps within each region into three gradations. The first includes values from 0 to 33 (Low), the second from 34 to 66 (Moderate) and the third from 67 and above (High)
Why did you choose to reduce the FPI values to a three-class scale? Was it, for example, to make inferences about causes of fire easier and faster than dealing with a continuous measure?
Lines 461-462 and equation 12: Therefore, we propose the following formula for partial verification of the fires received by Sentinel-2
Rather than a formula using set notation, it would be easier to simply say that fires were identified from the intersection of features from the Sentinel and Landsat data. Similarly, equation 13 can also be replaced by a few words. The set notation does not add anything and might puzzle some readers.
Lines 589-591: Having maps with a value from 0 to 100 is difficult to understand how potentially fire is possible for each region and to make some additional calculations. Therefore, the FPI was divided into three gradations Low, Moderate, High Fire danger.
Why is it difficult to understand? Please see also related comment above.
A minor amount of editing for English language usage would make the text more readable.
Author Response
Dear reviewer,
Thank you very much for your in-depth analysis of our work. Following your constructive comments and suggestions which helped to improve our study, we have made our best to address every point you raised. After taking careful attention to your comments and making a great improvement, we updated our manuscript and we expect you can consider our manuscript for publication.

Reviewer 3 Report (New Reviewer)
The manuscript is well-analyzed and requires some revision.
The north arrow and scalebar are missing in Fig 2a.
Line 422, how to determine the exact range of the FPI index?
I suggest adding links to the proposed GEE framework to facilitate the wider use of your method by the community.
Author Response
Dear reviewer,
Thank you very much for your in-depth analysis of our work. Following your constructive comments and suggestions which helped to improve our study, we have made our best to address every point you raised. After taking careful attention to your comments and making a great improvement, we updated our manuscript and we expect you can consider our manuscript for publication.

Round 2
Reviewer 1 Report (Previous Reviewer 3)
I appreciate the effort done by the authors to improve the paper according to my previous comments. Although many critical issues have been addressed in the revised manuscript, there are still some aspects of the study detailed below that need to be clarified by the authors. In the following my comments and suggestions:
Line 26-211: The introduction is too long and may be improved. Authors could introduce the topic of interest (content at lines 130-211) describing the limitations affecting the analysis of forest fires in Ukraine and the proposed solutions, while content at lines 30-121 may be included in a dedicated “Background” section.
Line 32: some characters are still in bold, please check.
Line 79: VIIRS aboard…
Line 106: Sentinel-2 MSI data up to 10 m spatial resolution of ..
Line 107: Fires at the early stage require the use of satellite data at very high temporal resolution (e.g., SEVIRI) to be promptly identified. On the other hand, Sentinel-2 MSI data may enable an efficient mapping of small fires when adequate detection methods are used. Please, change the sentence accordingly.
Line 118: proposed.
Line 181: MODIS and VIIRS data
Line 196: authors should provide more information about the Google Earth Engine platform, as already suggested during the previous revision process, citing papers (e.g., Gorelick et al., 2017) which describe in detail the GEE platform and its relative features
Line 203-204: this sentence should be smoothed.
Line 256: FIRMS is already defined at line 77.
Line 269: it seems that in 2021 a higher fire number occurred especially in the Northern region of Ukraine. This should be briefly discussed in the text.
Line 284: as in (1)
Line 289: The Multispectral Instrument (MSI) aboard Sentinel-2 satellites… ; please, specify the bands at different spatial resolution or add a table.
Line 307: While, OLI provides data at 30 m spatial resolution, TIRS…
Line 309: Data from the TIRS band 10 (….) resampled at…
Line 311: Since the combined revisit time of ..
Line 334: Sentinel-2 satellites…
Line 404: a scale bar and north arrow should be added to the map.
Line 481: Do you refer to brightness temperatures measured by TIRS in band 10? Please, specify this aspect in text and modify the sentence accordingly.
Line 486: Landsat 8/9 TIRS data…
Line 491: abnormal is less appropriated here.
Line 516: please specify which factors cause the spontaneous fire occurrence in Ukraine.
Line 525: you could refer to GEE here.
Line 545: burned areas (in red) retrieved from…
Line 546: burned mask (in red).. please, specify the bands used for the background images.
Line 550: Only a portion of the burned area seem to show temperature values above the selected threshold. The sentence should be modified accordingly. The use of TIRS data analyzed according to the proposed method does not represent in my opinion a valid solution to assess the results of this work. On the other hand, authors could use a proper detection method to detect active fires by means of Sentinel-2 MSI and Landsat 8/9 OLI data.
Line 553: see comment at line 491.
Line 572: Please specify the bands used to genarate the image displayed in background.
Line 595: GEE
Line 691-692: see comment at line 550. I suggest authors to reformulate the sentence to stress that other solutions need to be found to assess the information retrieved from maps of burned areas.
Line 693-694: this sentence is not clear and should be rewritten or removed.
some sentences have to be rewritten or modified.
Author Response
Dear reviewer,
Thank you very much for your analysis of our work. Following your comments and suggestions which helped to improve our study, we have made our best to address every point you raised. We updated our manuscript and we expect you can consider our manuscript for publication.

This manuscript is a resubmission of an earlier submission. The following is a list of the peer review reports and author responses from that submission.
Round 1
Reviewer 1 Report
This paper used remote sensing technology and a variety of satellite data to detect fires in Ukraine, and tried to analyze the causes of fires with the method of Fire Potential Index. The objectives of the study are clear, but there are some problems in the paper:
1. The methodology used in the paper is not innovative enough.
2. The explanation of the principle of FPI's algorithm is not sufficient enough to support the subsequent conclusions. The input parameters of FPI should be further analyzed, by which climatic, vegetation or other factors are these parameters affected, and whether the result of FPI truly reflect the probability of fire occurrence. Since this algorithm was originally used in the United States, its applicability in Ukraine needs also be discussed. It is necessary to give the exact values of the correlation between real fire locations and FPI data for the past several years.
3. In order to ensure the rigor of the conclusions, the paper should indicate more comprehensively what factors influence the occurrence of fires and whether these factors are all taken into account in the FPI calculation process. If there are significant deviations between the result of FPI and actual burn area, how to exclude all the other causes such as lightning strikes, the other human activities except military activity and so on.
In summary, the study of fire detection and the exploration of their causes is relevant. But the innovation and rigor of this paper are somewhat deficient.
The English expressions are smoothly understandable, but there are still some sentences that are not clearly formulated and need to be revised.
Reviewer 2 Report
The paper deals with using satellite data to monitor forest fires occurred in Ukraine during the war. Using satellite data for this kind of application is a well-known practice, therefore the proposed idea is not a novelty.
Authors integrated different satellite data, with different capabilities with regards fire detection, to assess the potential of their work, and this makes the approach confusing.
FIRMS data are based on the capability of sensors like MODIS and VIIRS to detect high brightness temperatures in the Medium Infrared, when a fire is in progress. Visible Sentinel 2 data and Thermal Landsat 8-9 data were used by the authors just to look for burned areas in the area already detected as affected by fires. What is the novelty here?
Why did the authors not use VIIRS-FIRMS data at 375 m of spatial resolution?
Why did the authors not investigate the potential of Sentinel 2 and Landsat 8-9 NIR and SWIR data to infer fire presence?
Finally, the answers to the two scientific questions they posed are intrinsically related to the used scheme, because selecting just a subset of Ukraine's territory has forced their results, as well as investigating the area more affected by war has already indicated it as the main cause of fires in 2022.
It is not true that there is no way to manage the limitation due to using fixed thresholds on satellite signals. Multi-temporal approaches can provide more dynamic thresholds that can help in reducing the sensitivity/accuracy issues due to fixed values.
Reviewer 3 Report
The paper presents the results of the fire monitoring in Ukraine performed through satellite data at different spatial resolution. Although the topic may be certainty of interest for the readers, the manuscript requires a significant revision. In detail, the abstract is quite confusing and needs to be rewritten. The introduction is confusing too, and there is no mention of previous papers focusing on the same topic (fires caused by the war in Ukraine). Consequently, it is difficult to evaluate the originality of the study; moreover, in the introduction authors make confusion among satellite, sensors and systems. Some paragraphs of the data section need to be included in the methods section (e.g., FIRMS; FPI). In addition, the methodological approach is questionable (e.g., use of Landsat 8/9 TIRS data to validate fire products at higher spatial resolution). In the discussion section, factors favoring the spontaneous fire occurrence in the region of interest are not analysed in detail by the authors, and the impact of meteorological clouds on results of the study is not quantified. Therefore, I do not recommend the paper for publication.